# Prefrontal cortical ChAT-VIP interneurons provide local excitation by cholinergic synaptic transmission and control attention

Joshua Obermayer[1,5], Antonio Luchicchi[1,3,5], Tim S. Heistek[1], Sybren F. de Kloet[1], Huub Terra[1], Bastiaan Bruinsma[1], Ouissame Mnie-Filali[1], Christian Kortleven[1], Anna A. Galakhova [1], Ayoub J. Khalil[1], Tim Kroon [1,4], Allert J. Jonker[2], Roel de Haan [1], Wilma D.J. van den Berg [2], Natalia A. Goriounova [1], Christiaan P.J. de Kock[1], Tommy Pattij [2]* & Huibert D. Mansvelder [1]*

Neocortical choline acetyltransferase (ChAT)-expressing interneurons are a subclass of vasoactive intestinal peptide (ChAT-VIP) neurons of which circuit and behavioural function are unknown. Here, we show that ChAT-VIP neurons directly excite neighbouring neurons in several layers through fast synaptic transmission of acetylcholine (ACh) in rodent medial prefrontal cortex (mPFC). Both interneurons in layers (L)1–3 as well as pyramidal neurons in L2/3 and L6 receive direct inputs from ChAT-VIP neurons mediated by fast cholinergic transmission. A fraction (10–20%) of postsynaptic neurons that received cholinergic input from ChAT-VIP interneurons also received GABAergic input from these neurons. In contrast to regular VIP interneurons, ChAT-VIP neurons did not disinhibit pyramidal neurons. Finally, we show that activity of these neurons is relevant for behaviour and they control attention behaviour distinctly from basal forebrain ACh inputs. Thus, ChAT-VIP neurons are a local source of cortical ACh that directly excite neurons throughout cortical layers and contribute to attention.

[1] Department of Integrative Neurophysiology, Center for Neurogenomics and Cognitive Research (CNCR), Vrije Universiteit, Amsterdam Neuroscience, The Netherlands. [2] Department of Anatomy and Neurosciences, Amsterdam UMC, Vrije Universiteit, Amsterdam Neuroscience, The Netherlands. [3] Present address: Department of Anatomy and Neurosciences, Clinical Neuroscience, Amsterdam UMC, Vrije Universiteit, Amsterdam Neuroscience, The Netherlands. [4] Present address: MRC Centre-Developmental Neurobiology, King's College London, London, UK. [5] These authors contributed equally: Joshua Obermayer, Antonio Luchicchi [6] These authors jointly supervised this work: Tommy Pattij, Huibert D. Mansvelder  *email: t.pattij@vumc.nl; h.d.mansvelder@vu.nl

The neurotransmitter acetylcholine (ACh) shapes activity of cortical neurons and supports cognitive functions such as learning, memory and attention[1–3]. Rapid ACh concentration changes in rodent medial prefrontal cortex (mPFC) occur during successful stimulus detection in a sustained attention task[4,5]. Traditionally, it is assumed that neocortical ACh is released exclusively from terminals of axonal projections whose cell bodies reside in basal forebrain (BF) nuclei[6,7]. Chemical lesions of cholinergic BF projections impair attention behaviour[8–12] and optogenetic activation of BF cholinergic neurons can mimic ACh concentration changes typically observed during attention behaviour[11]. However, well over 30 years ago, interneurons intrinsic to the neocortex have been described that express the ACh-synthesizing enzyme choline acetyltransferase (ChAT)[13–15]. Whether these neurons are a local source of ACh in the prefrontal cortex relevant for attention is not known.

ChAT-expressing interneurons are a sparse population of about 1% of all cortical neurons, which are more abundantly present in superficial cortical layers 2/3 (L2/3) than in deep layers[16]. They have either a bipolar or multipolar morphology and express vasoactive intestinal peptide (VIP)[13,14,17]. In mouse cortex, about 15% of VIP neurons express ChAT[18], while in the PFC of rats about 30% of VIP neurons express ChAT[16]. Indirect evidence suggested that these ChAT-VIP neurons release ACh in neocortical circuits upon activation, inducing an increase in spontaneous glutamatergic synaptic transmission received by L2/3 pyramidal neurons[19]. Despite molecular, morphological and physiological characterizations, technical limitations thus far prevented a direct demonstration of whether these ChAT-expressing VIP interneurons release ACh. Moreover, basal forebrain cholinergic neurons that project to the neocortex have been shown to form direct point-to-point synapses with several types of neurons in different layers, thereby modulating their activity on a millisecond time scale[20–24]. Activation of ChAT-VIP interneurons can slowly alter local synaptic activity[19], but it is unknown whether ChAT-VIP interneurons do this via direct cholinergic synaptic transmission, or whether they modulate local neuronal activity more diffusely.

Neocortical circuits contain distinct classes of interneurons with characteristic innervation patterns of local cortical neurons[25–28]. Fast-spiking (FS), parvalbumin-expressing (PV) interneurons perisomatically innervate pyramidal neurons, while low-threshold spiking (LTS), somatostatin-expressing (SST) genes target more distal regions of dendrites[29]. GABAergic VIP neurons inhibit PV and SST interneurons, thereby disinhibiting pyramidal neurons[28,30,31]. Single cell transcriptomic analysis of cortical neurons has shown that distinct subtypes of VIP interneurons exist with unique gene expression profiles[18,25]. Whether VIP interneuron subtypes are functionally distinct is not known[32]. It is also not known whether ChAT-expressing VIP interneurons show similar innervation patterns, specifically targeting neighbouring PV and SST interneurons, and activating disinhibitory pathways.

Here, we address these issues and focus in particular on elucidating how local neuronal circuitry in various layers of the mPFC are affected by ChAT-VIP neuron activation. We find that ChAT-VIP interneurons directly excite local interneurons and pyramidal neurons in different mPFC layers with fast cholinergic synaptic transmission. In addition, we show that despite their sparseness, activity of ChAT-VIP neurons is involved in sustained attentional performance in freely moving animals in a manner distinct from basal forebrain cholinergic inputs to the prefrontal cortex.

## Results

### Fast cholinergic synaptic transmission by ChAT-VIP neurons.
Previous studies in mice have shown that activation of ChAT-VIP

interneurons increases spontaneous excitatory postsynaptic potentials (EPSPs) in layer 5 pyramidal neurons[19]. However, it is unresolved whether ChAT-VIP interneurons directly innervate other neurons in the cortex. To address this, we first expressed channelrhodopsin-2 (ChR2) in ChAT-VIP interneurons in the mPFC of ChAT-cre mice[33] (Fig. 1a) and recorded from L1 interneurons since these neurons are known to reliably express nicotinic acetylcholine receptors (nAChRs) in other neocortical areas[34–36]. In mouse mPFC, L1 interneurons received miniature excitatory postsynaptic currents (mEPSCs) mediated by nAChRs (Supplementary Fig. 1). All brain slice physiology experiments in this study were done in the presence of glutamate receptor blockers (DNQX, 10 μM; AP5, 25 μM). We made simultaneous whole-cell patch-clamp recordings of EYFP-positive ChAT-VIP neurons in L2/3 and nearby L1 interneurons in mouse mPFC (Fig. 1b). EYFP-positive neurons showed similar morphology, ChAT, VIP, CR, GAD expression patterns (Fig. 1a; Supplementary Figs. 2 and 3) and action potential (AP) profiles (Fig. 1c) as was reported previously[14,16,19]. First, we triggered single APs in presynaptic ChAT-VIP interneurons by short (1 ms) electrical depolarization of the membrane potential (Fig. 1d, $n = 11$). This resulted in inward currents in postsynaptic L1 interneurons that either were fast and lasted up to 10 ms (Fig. 1d, top traces), or had slower kinetics and lasted over 100 ms (Fig. 1d, bottom traces). The fast postsynaptic currents were fully blocked by methyllycaconitine (MLA, 100 nM), an antagonist of α7-subunit-containing nAChRs (Fig. 1d, grey trace, Fig. 1j). The slow postsynaptic current was fully blocked by DHßE (10 μM), an antagonist of β2-subunit-containing nAChRs (Fig. 1d, bottom, grey trace, Fig. 1j). Postsynaptic currents occurred time-locked to presynaptic APs with onset delays of $1.26 \pm 0.02$ ms (average ± sem) from the peak of the AP (Fig. 1e), suggesting synaptic transmission. Postsynaptic currents showed substantial amplitude fluctuations, in line with synaptic transmission, with an average amplitude of $-30 \pm 15$ pA (Fig. 1f, j). Synaptic transmission between ChAT-VIP and L1 interneuron was quite reliable: the majority of unitary synapses transmitted ACh at each presynaptic AP and average failure rates were $35 \pm 10\%$ (Fig. 1f, k). Single cell mRNA expression data from the Allen Institute for Brain Science database on mouse cell types[18,25] shows that ChAT-VIP neurons express the vesicular acetylcholine transporter (VAChT) encoded by the SLC18A3 gene (Supplementary Fig. 3E). Thus, ACh is likely released by means of vesicles from these neurons.

Next, in the same recordings, we induced APs in presynaptic ChAT-VIP neurons by activating ChR2 with brief blue-light pulses (Fig. 1g). This induced similar inward currents in postsynaptic L1 interneurons with similar current amplitudes ($-39 \pm 20$ pA; Fig. 1g, i, j) and that were blocked by the nicotinic receptor antagonists MLA and DHßE (Fig. 1g, j). The distribution of synaptic onset delays, calculated from the peak of the presynaptic AP, was significantly shorter than with electrically-induced APs ($0.68 \pm 0.02$ ms; Mann–Whitney test $P < 0.00001$; Fig. 1h). This lower synaptic onset delay with optogenetic stimulation could result from more rapid activation of additional ChR2-expressing ChAT-VIP neurons in the slice that innervate the same L1 interneuron, or from direct depolarization of presynaptic terminals by blue-light-mediated ChR2 activation, or both. Failure rates of postsynaptic current induction and postsynaptic current amplitude distribution in these recordings were similar with electrical and light-induced presynaptic AP induction (Fig. 1i–k). To test how abundant ChAT-VIP input to L1 interneurons is in mouse prefrontal cortical slices, we recorded from an additional set of 118 L1 interneurons and activated ChAT-VIP neurons with blue-light pulses. Overall, we found that 36% of L1 interneurons received inputs from ChAT-VIP neurons ($n = 42$ of 118 neurons, Fig. 1l) and these were all

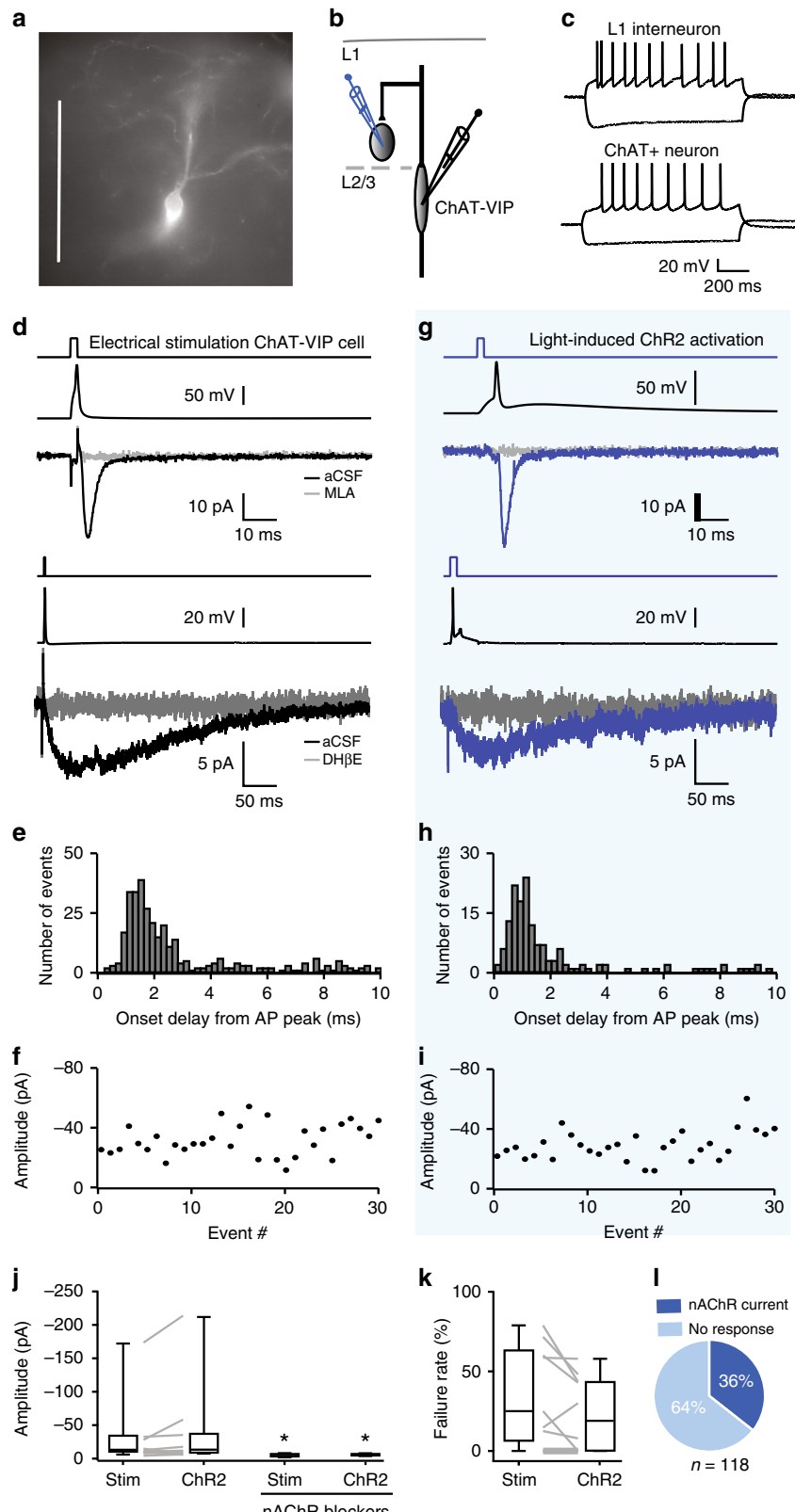

blocked by a cocktail of nicotinic AChR blockers (MLA and DHβE). Thus, in mouse mPFC, ChAT-VIP interneurons provide fast cholinergic synaptic input to L1 interneurons mediated by nicotinic AChRs.

ChAT-VIP neurons are more abundant in the mPFC of rats than in mouse cortex[16,18]. To test whether ChAT-VIP neurons

more reliably innervate L1 interneurons in rat neocortex, we expressed ChR2 in ChAT-VIP interneurons in the mPFC of ChAT-cre rats[37]. Following mPFC injections, we did not observe significant retrograde labelling of cells in the basal forebrain (Supplementary Fig. 4A, B). In rat prefrontal cortex, EYFP-positive L2/3 ChAT-VIP neurons also had a bipolar

**Fig. 1** Fast cholinergic synaptic transmission by ChAT-VIP cells. **a** EYFP-labelled ChAT-VIP interneuron targeted for recording in an acute mouse mPFC brain slice (DIC-IR microscope). Scale bar 100 μm. **b** Schematic of simultaneous recordings of presynaptic ChAT-VIP interneurons and postsynaptic L1 interneurons. **c** Membrane potential responses of a L1 interneuron (top) and L2/3 ChAT-VIP interneuron (bottom) to depolarizing (+200 pA) and hyperpolarizing (−150 pA) somatic current injection. **d** Example traces of a synaptically-connected ChAT-VIP cell and a L1 interneuron. Top trace: short step depolarization. Second trace: Presynaptic AP in ChAT-VIP interneuron. Third trace: fast postsynaptic inward current of the L1 interneuron (black trace, average of 45 sweeps), blocked by nAChR antagonists (MLA 100 nM, grey trace). Fourth trace: short step depolarization. Fifth trace: Presynaptic AP in ChAT-VIP interneuron. Bottom trace: slow postsynaptic inward current of the L1 interneuron (black trace, average of 57 sweeps), blocked by nAChR antagonists (DHßE 10 μM, grey trace). **e** Histogram of onset delays of postsynaptic currents relative to the peak amplitude of the electrical stimulation-induced presynaptic AP (from n = 11 connected cell pairs). **f** Amplitudes of successive unitary synaptic currents in the same ChAT-VIP L1-interneuron cell pair as in (**d**). **g** Recordings from the same neuron pair as in (**d**), but now AP firing was induced by activating ChR2 with a blue-light flash. Traces and graph as in (**c**). Here and in all other figures, for all example traces of responses to optogenetic stimulations 20 sweeps were averaged. **h** Histogram of onset delays of postsynaptic currents relative to the peak amplitude of the ChR2-induced presynaptic AP (from n = 11 connected cell pairs). **i** Amplitudes of successive unitary synaptic currents in the recording of the same ChAT-VIP L1-interneuron cell pair in (**f**), but with ChR2-induced presynaptic APs. **j** Summary of postsynaptic current amplitudes. Box plot centre line: median; bounds of box: 25th to 75th percentile; whiskers: min-max data bound. **k** Summary of failure rates in the same 11 L1 interneurons. **l** Postsynaptic response incidence in a separate set of 118 L1 interneurons following ChR2-induced ChAT-VIP interneuron activation.

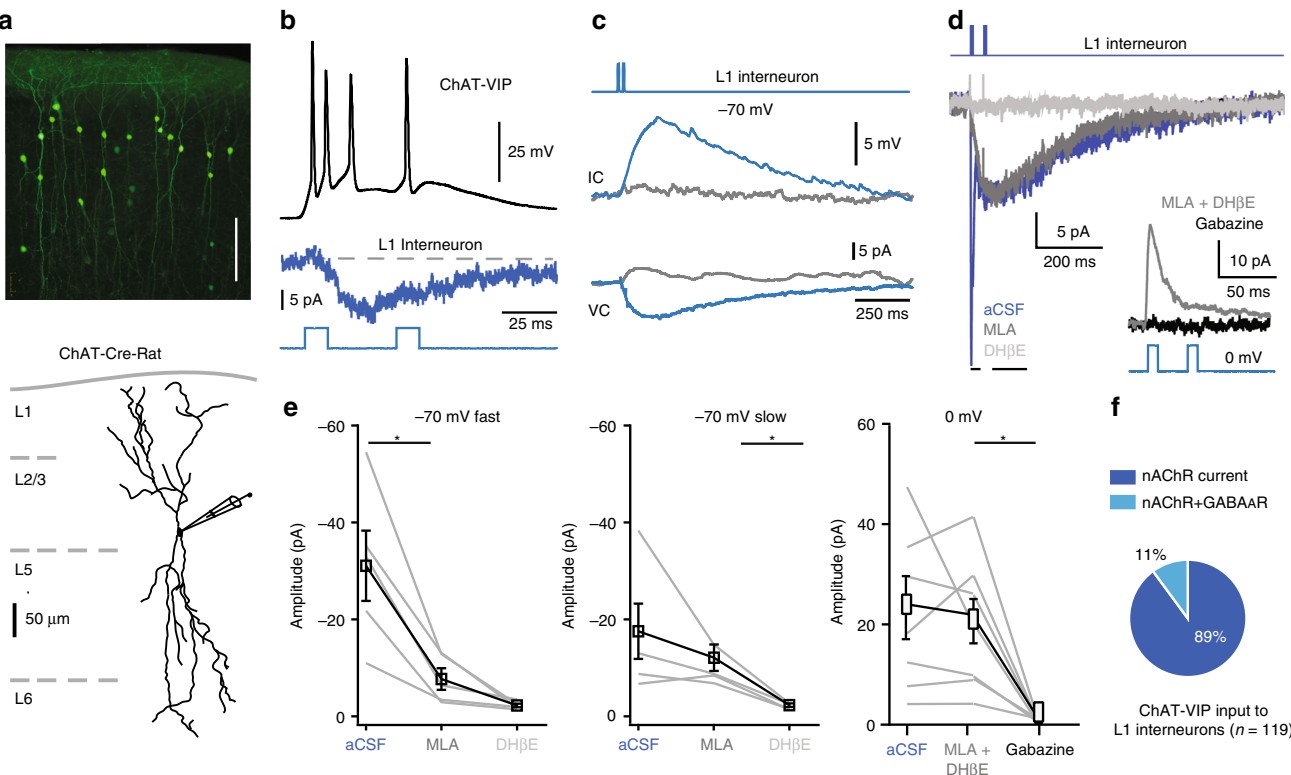

**Fig. 2** ChAT-VIP interneurons release ACh and GABA. **a** EYFP-labelled ChAT-VIP interneurons in rat mPFC. Labelled cells in L2/3 have bipolar morphology (scale bar 200 μm). Below: Digital reconstruction of an EYFP-positive ChAT-VIP interneuron in the rat mPFC. **b** Response to blue-light-induced ChR2 activation (470 nm, 10 ms, 25 Hz) of a rat mPFC ChAT-VIP neuron (black trace, top panel, voltage response). Middle blue trace: postsynaptic response in a simultaneously recorded L1 interneuron. Bottom: blue-light stimulation protocol applied. **c** L1 interneuron is depolarized by blue-light ChR2-mediated activation of ChAT-VIP cells. A single component inward current underlies the depolarization. Both the inward current and the depolarization are blocked by nAChR blockers (MLA 100 nM and MEC 10 μM, grey traces). **d** L1 interneuron recording showing light-evoked biphasic synaptic input currents at −70 mV in aCSF (blue trace). In the presence of MLA (100 nM), the fast component is selectively blocked (dark grey trace). Adding DHßE (10 μM) blocked the slow component (light grey trace). Inset: L1 interneuron recording at 0 mV showing light-evoked synaptic currents in the presence of nAChR blockers (grey trace) or gabazine (10 μM, black trace). **e** Summary plots of pharmacology of postsynaptic responses in L1 interneurons in response to light-induced ChR-mediated activation of ChAT-VIP neurons. Left panel: amplitudes of fast postsynaptic currents at −70 mV in aCSF and with nAChR blockers MLA and DHßE (aCSF: −30.92 ± 7.24 pA, MLA: −7.68 ± 2.21 pA, p = 0.014, paired t-test, two-tailed, t = 3.888, df = 4, n = 5). Middle panel: amplitudes of slow postsynaptic currents at −70 mV in aCSF and with nAChR blockers MLA and DHßE (aCSF: −17.47 ± 5.71 pA, DHßE: −2.31 ± 0.40 pA, p = 0.015, paired t-test, two-tailed, t = 3.888, df = 4, n = 5). Right panel: amplitudes recorded at 0 mV (aCSF: 20.18 ± 5.794 pA, nAChR blockers: 22.77 ± 7.932 pA, gabazine: 1.803 ± 0.6177 pA, one-way ANOVA: $F_{(6,12)}$ = 2.256, p = 0.0220, n = 7). Summary plots show averages ± SEM. **f** Incidence of L1 interneurons in rat mPFC receiving nAChR and GABA$_A$R-mediated synaptic inputs from ChAT-VIP interneurons.

morphological appearance (Fig. 2a), as was originally reported[14,16], confirming their identity. Upon activation of ChR2, ChAT-VIP interneurons fired APs and simultaneously recorded L1 interneurons showed postsynaptic inward currents (Fig. 2b). In all recorded L1 interneurons ($n = 119$), blue-light activation of ChR2 expressed by ChAT-VIP neurons generated postsynaptic depolarizations and inward currents that were blocked by nAChR blockers (Fig. 2c). These currents were either mono-phasic, consisting of only a fast or slow component, or were biphasic, consisting of a fast and a slow component (Fig. 2d), reminiscent of synaptic fast α7-containing nAChR and slow β2-containing nAChR currents observed in Fig. 1d, g and expressed by L1 interneurons in sensory cortical areas[34–36]. Indeed, the fast current component was selectively blocked by MLA, while the slow component was selectively blocked by DHßE (Fig. 2d, e).

Since ChAT-VIP interneurons can co-express the acetylcholine (ACh) synthesizing enzyme ChAT and the GABA synthesizing enzyme GAD (Supplementary Fig. 3)[16,19], we asked whether these neurons release GABA in addition to ACh. To test this, the membrane potential of rat mPFC L1 interneurons was held at 0 mV in the presence of nAChR blockers (Fig. 2d inset). Blue-light activation of ChR2-expressing ChAT-VIP cells evoked fast outward currents in 11% of the cells ($n = 13/119$), which were blocked by gabazine (10 μM; Fig. 2d, e). Thus, while all L1 interneurons in rat mPFC received fast cholinergic inputs from ChAT-VIP cells mediated by nAChRs, a minority of L1 neurons also received GABA from ChAT-VIP neurons (Fig. 2f). In all figures below (Figs. 3–6), data from rat mPFC are shown.

**Direct excitation, but no disinhibition by ChAT-VIP neurons.** VIP interneurons have been shown to disinhibit L2/3 pyramidal neurons by inhibiting activity of fast-spiking, PV-expressing interneurons and low-threshold spiking, SST-expressing interneurons[28,30,31]. To test whether ChAT-VIP interneurons form disinhibitory circuits in L2/3, we made whole-cell patch-clamp recordings of L2/3 pyramidal neurons in rat mPFC and triggered activity in ChR2-expressing ChAT-VIP interneurons by applying blue-light pulses. Light-induced activation of ChAT-VIP interneurons did not induce GABAergic synaptic currents in pyramidal neurons, but induced depolarizing inward currents in a minority of pyramidal neurons ($n = 3$ of 18; Fig. 3a). These inward currents were blocked by nAChR blockers DHßE and MLA. Next, we analysed spontaneous inhibitory postsynaptic currents (sIPSCs) received by L2/3 pyramidal neurons. Light-induced activation of ChAT-VIP interneurons did not alter the frequency of sIPSCs received by L2/3 pyramidal neurons (Fig. 3b, c), indicating that activity of ChAT-VIP neurons did not change inhibition received by L2/3 pyramidal neurons. To test whether ChAT-VIP neurons target and inhibit other local interneuron types, we recorded from rat mPFC FS (Fig. 3d) and LTS (Fig. 3f) interneurons, as defined by their AP firing profiles, while triggering activity in ChR2-expressing ChAT-VIP interneurons by applying blue-light pulses (Fig. 3e, g). We did not observe any GABA-mediated inhibitory postsynaptic currents at 0 mV membrane potential in both interneuron types following activation of ChAT-VIP interneurons (Fig. 3e, g). However, a subgroup of FS ($n = 4/6$) as well as LTS ($n = 5/8$) interneurons showed small inward currents at −70 mV that were mediated by fast α7-containing or slow β2-containing nAChRs and were blocked after application of the nAChR antagonists DHßE and MLA (Fig. 3e, g). Thus, we find no evidence that ChAT-VIP neurons form disinhibitory circuits in L2/3, as has been reported for other VIP interneurons, but we do find evidence that ChAT-VIP interneurons directly excite subgroups of local interneurons as well as a minority of L2/3 pyramidal neurons.

**Cholinergic synaptic inputs to L6 pyramidal neurons.** Previous studies have shown that a majority of layer 6 pyramidal neurons express nAChRs[38,39] and these neurons can be activated by cholinergic inputs from the BF[21,22,40]. EYFP-positive fibres of ChAT-VIP cells show abundant arborization in deep layers of rat mPFC (Fig. 4a; Supplementary Fig. 2). We therefore asked whether L6 pyramidal neurons receive direct inputs from ChAT-VIP interneurons. To test this, we made whole-cell patch-clamp recordings from rat mPFC L6 pyramidal neurons combined with activation of ChR2-expressing ChAT-VIP interneurons by applying blue-light pulses (Fig. 4b, c). Seventy-one percent ($n = 20/28$) of recorded L6 pyramidal neurons showed nAChR antagonist sensitive inward currents with slow kinetics (Fig. 4b, d). Although the amplitude of these currents was on average about 5 pA, the currents lasted about 1 s and ChAT-VIP activation resulted in a significant depolarization of the membrane potential (Fig. 4b, c) due to the relatively high membrane resistance of these cells[22,34]. Six of the L6 pyramidal cells showed an additional gabazine-sensitive fast outward current at 0 mV in the presence of nAChR blockers (Fig. 4c, d). These findings show that 20 out of 28 L6 pyramidal neurons receive direct cholinergic inputs from local ChAT-VIP interneurons, and 6 out of 28 L6 pyramidal neurons received both ACh and GABA (Fig. 4e).

**Consequences of co-transmission of ACh and GABA.** ChAT-VIP cells activate postsynaptic nAChRs that depolarize pyramidal neurons and interneurons in L1, L2/3 and L6. It is somewhat surprising that some of these postsynaptic neurons also receive GABA and show inhibitory GABAR currents. It is not known what the consequences for the excitability of postsynaptic cells is of co-transmission of GABA and ACh. Typically, GABAR and nAChR currents had distinct kinetics (Fig. 5a, b). The majority of excitatory nAChR-mediated synaptic responses had slow kinetics with rise times of 155.5 ± 26.5 ms ($n = 9/13$ postsynaptic L1 cells with combined GABAR and nAChR currents; Fig. 5a, c, d). GABAergic currents had much faster kinetics that decayed back to baseline in about 30 ms (Fig. 5b, d). In only four postsynaptic L1 interneurons that showed a combined nAChR and GABAR-mediated currents we found both the fast MLA-sensitive nAChR current that would match the activation kinetics of the fast GABAR current ($n = 4/13$). Thus, in the majority of combined GABA and ACh responses, fast GABAR currents were followed by slower nAChR currents.

Activation of GABAR currents could lead to shunting inhibition, preventing or postponing subsequent AP firing. Alternatively, hyperpolarizing GABAergic inputs can give rise to rebound excitation by deinactivation of intrinsic voltage-gated conductances[41–43]. The excitation induced by slow inward nAChR currents may thus theoretically be amplified by rebound excitation following the GABAergic hyperpolarization. To test whether this occurs, we recorded from L1 interneurons and monitored AP timing in response to ramp depolarizations with and without blue-light activation of ChR2-expressing ChAT-VIP interneurons in rat mPFC. First, to test the effect of the cholinergic component of ChAT-VIP input, only recordings with nAChR-mediated postsynaptic currents without GABAR currents were included (Fig. 5e). Activation by blue-light pulses of ChR2-expressing ChAT-VIP interneurons advanced the timing of the first AP during ramp depolarizations (Fig. 5f), reducing the AP onset delay (Fig. 5i). Next, we investigated whether co-transmission of GABA advances or postpones the first AP. Now, only recordings with combined nAChR/GABAR-mediated post-synaptic currents were included (Fig. 5g). Blocking GABAergic inhibition with the GABA$_A$ receptor antagonist gabazine resulted in a shortening of the delay to the first AP in L1 interneurons

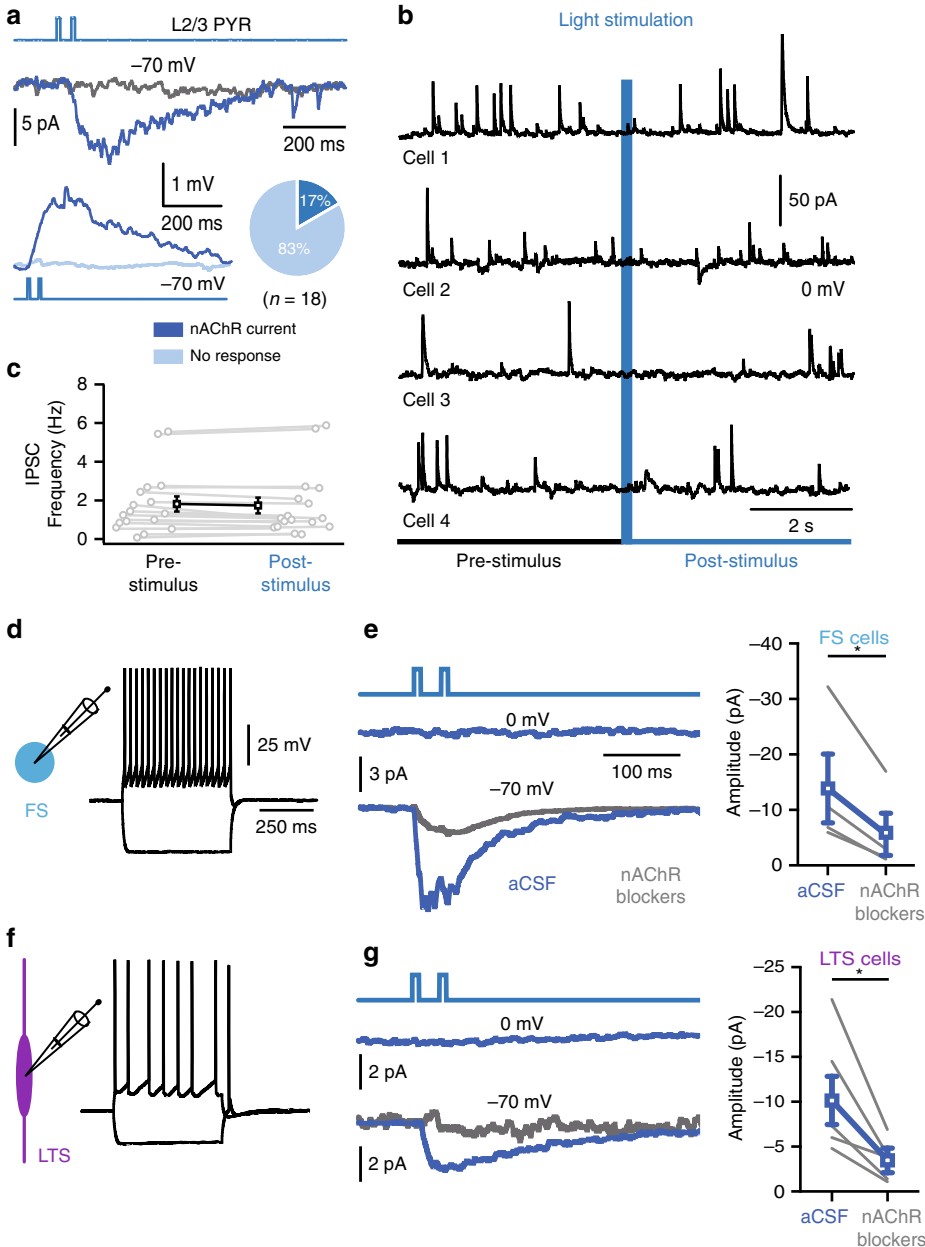

**Fig. 3** Direct excitation, but no disinhibition by ChAT-VIP neurons. **a** Example traces of postsynaptic responses in a rat L2/3 pyramidal neuron upon ChR2-mediated activation of ChAT-VIP interneurons with blue light (470 nm, 10 ms, 25 Hz, top trace). The postsynaptic responses were blocked by nAChR antagonists (MLA 100 nM and MEC 10 μM, grey trace). The majority of L2/3 pyramidal neurons did not show a response to ChAT-VIP neuron activation (light blue trace). Pie chart showing the percentages of L2/3 pyramidal neurons with nAChR-mediated postsynaptic response (dark blue) and without (light blue). **b** Example traces of L2/3 pyramidal neurons recorded at 0 mV receiving spontaneous GABAergic IPSCs. Five blue-light pulses were applied (25 Hz, 10 ms). **c** Comparison of spontaneous IPSC frequency in L2/3 pyramidal neurons before and after ChR2-mediated activation of ChAT-VIP interneurons with five blue-light pulses (25 Hz, 10 ms), (IPSC frequency pre-stimulus: 1.809 ± 0.389 Hz, post stimulus: 1.732 ± 0.411 Hz, $p = 0.2310$, paired $t$-test, two-tailed, $t = 1.245$, $df = 16$; $n = 17$; mean ± SEM). **d** AP profile of a rat mPFC L2/3 FS interneuron in response to somatic step current injection (+200 pA and −150 pA). **e** Left: Example traces (average of single 10 traces) of postsynaptic responses in an FS interneuron upon ChAT-VIP interneurons with blue light (470 nm, 10 ms, 25 Hz, top trace). Middle trace: example trace recorded at 0 mV showing absence of an IPSC ($n = 0/6$). Bottom traces: light-evoked postsynaptic currents ($n = 4/6$) in absence (blue trace) or presence of nAChR blockers (MLA 100 nM and MEC 10 μM, grey trace). Right: Summary plot of the postsynaptic current amplitudes and blocked by nAChR blockers. **f** AP profile of a L2/3 LTS interneuron. **g** As in (**e**) but for a rat mPFC L2/3 LTS interneuron. No GABAergic IPSCs at 0 mV were observed following light-evoked activation of ChAT-VIP interneurons ($n = 0/8$). A subgroup of LTS neurons showed light-evoked postsynaptic currents ($n = 5/8$) at −70 mV (blue trace) that was blocked by nAChR antagonists (MLA 100 nM and MEC 10 μM, grey trace). Right: Summary plot of the postsynaptic current amplitudes. Summary plots show averages ± SEM.

during ramp depolarizations (Fig. 5h, i), suggesting that the postsynaptic GABAR currents provided shunting inhibition that postponed AP firing. Gabazine did not alter the input resistance (Fig. 5i inset), nor excitability of L1 interneurons and did not

advance spiking in L1 neurons that did not show co-transmission of GABA (Supplementary Fig. 5F). In line with these findings, at near-AP threshold membrane potentials in L1 interneurons, blue-light activation of ChR2-expressing ChAT-VIP interneurons

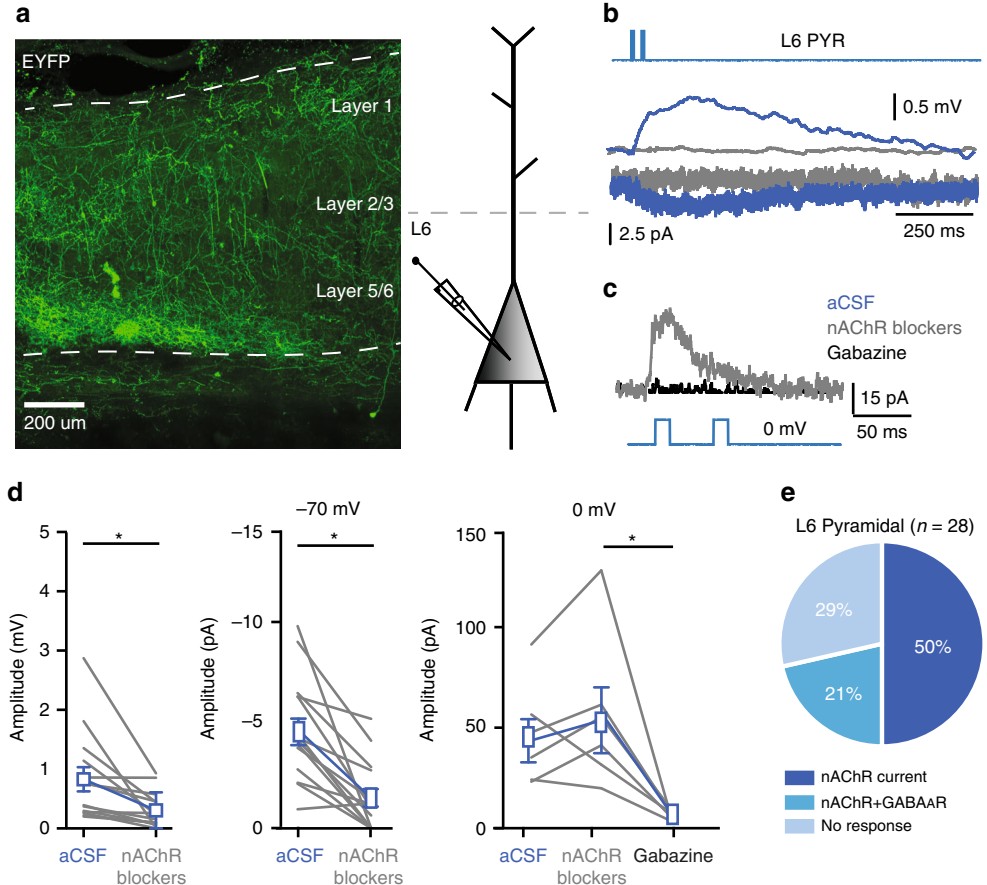

**Fig. 4** Direct synaptic inputs to L6 pyramidal neurons. **a** EYFP expression in ChAT-VIP neurons throughout mPFC layers. Right: schematic illustration of recording set up. **b** Example traces from a rat L6 pyramidal neuron showing depolarization and an inward current at −70 mV in response to blue-light ChR2-mediated activation of ChAT-VIP neurons (470 nm, 10 ms, 25 Hz) in absence (blue trace) or in the presence of nAChR antagonists (grey trace). **c** Same L6 pyramidal neuron recorded at 0 mV membrane potential showing light-evoked synaptic current in the presence of nAChR blockers (grey trace) and gabazine (black trace). **d** Left: summary chart showing the amplitudes of postsynaptic membrane potential changes with and without nAChR blockers. Middle: postsynaptic current amplitudes at −70 mV membrane potential without and with nAChR blockers (aCSF: 4.820 ± 0.6853 pA, nAChR blockers: 1.483 ± 0.4594 pA, $p = 0.0002$, paired $t$-test, two-tailed, $t = 5.051$, $df = 13$; $n = 14$, mean ± SEM). Right: postsynaptic current amplitudes recorded at 0 mV with nAChR blockers and gabazine (aCSF: 40.85 ± 10.35 pA, nAChR blockers: 50.65 ± 15.47 pA, gabazine: 1.403 ± 0.8461 pA, one-way ANOVA: $F_{(5,10)} = 2.949$, $p = 0.0148$; $n = 6$, mean ± SEM). Summary plots show averages ± SEM. **e** Pie chart showing percentages of L6 pyramidal neurons with nAChR-mediated, combined nAChR and GABA$_A$R-mediated, and no synaptic currents.

augmented AP firing probability much more when GABARs were blocked by gabazine (Supplementary Fig. 5A–E). Taken together, these results show that postsynaptic nAChR currents induced by ChAT-VIP interneurons directly excited L1 interneurons, increasing AP firing probability and shortening delays to first AP firing. Co-transmission of GABA provided shunting inhibition, postponing AP firing, rather than facilitating rebound excitation.

**ChAT-VIP neurons are required for attention.** Is activity of ChAT-VIP interneurons relevant for mPFC function during behaviour? To test this, we optogenetically inhibited ChAT-VIP cells during a well-validated task for assessing attention behaviour in rodents, the 5 choice serial reaction time task (5CSRTT)[44] (Fig. 6; Supplementary Fig 6). In this task, animals observe a curved wall with 5 holes in an operant chamber. After an inter-trial interval, typically 5 s, a cue light will briefly turn on for 1 s, in a randomly assigned hole. Following cue light presentation, animals have to respond within 2 s with a nose poke in the illuminated hole, to obtain a food reward (Fig. 6b). During a daily session, animals complete 100 trials. Since ChAT-VIP cells release

ACh in the mPFC, similar to basal forebrain (BF) cholinergic inputs, we also tested whether inhibiting ChAT-VIP interneurons affected attention behaviour distinct from inhibiting BF choli-nergic inputs to mPFC. Therefore, ChAT::cre rats received AAV5::DIO-EYFP-ARCH3.0 (or AAV5::DIO-EYFP in controls) injections either in the mPFC or the BF and optic fibres were placed over the prelimbic (PrL) mPFC in all groups (Fig. 6a; Supplementary Figs. 4 and 6). By randomly assigning half of the trials to green laser light ON and the other half to laser OFF (50 trials each, randomly mixed ON/OFF), ChAT-VIP cells or BF-to-mPFC projections were either free to fire APs or were inhibited in the same animals for five seconds during the pre-cue period when rats show preparatory attention for the upcoming stimulus presentation (Fig. 6b). For each animal, behavioural performance during laser-ON trials was compared with its own behavioural performance during laser-OFF trials. Inhibiting ChAT-VIP cells or BF-to-mPFC projections impaired response accuracy (Fig. 6c), and both inhibition of BF cholinergic neurons as well as inhibition of ChAT-VIP inter-neurons reduced correct responses and increased errors in each animal (Fig. 6d, e). Interestingly, no changes in any of the other

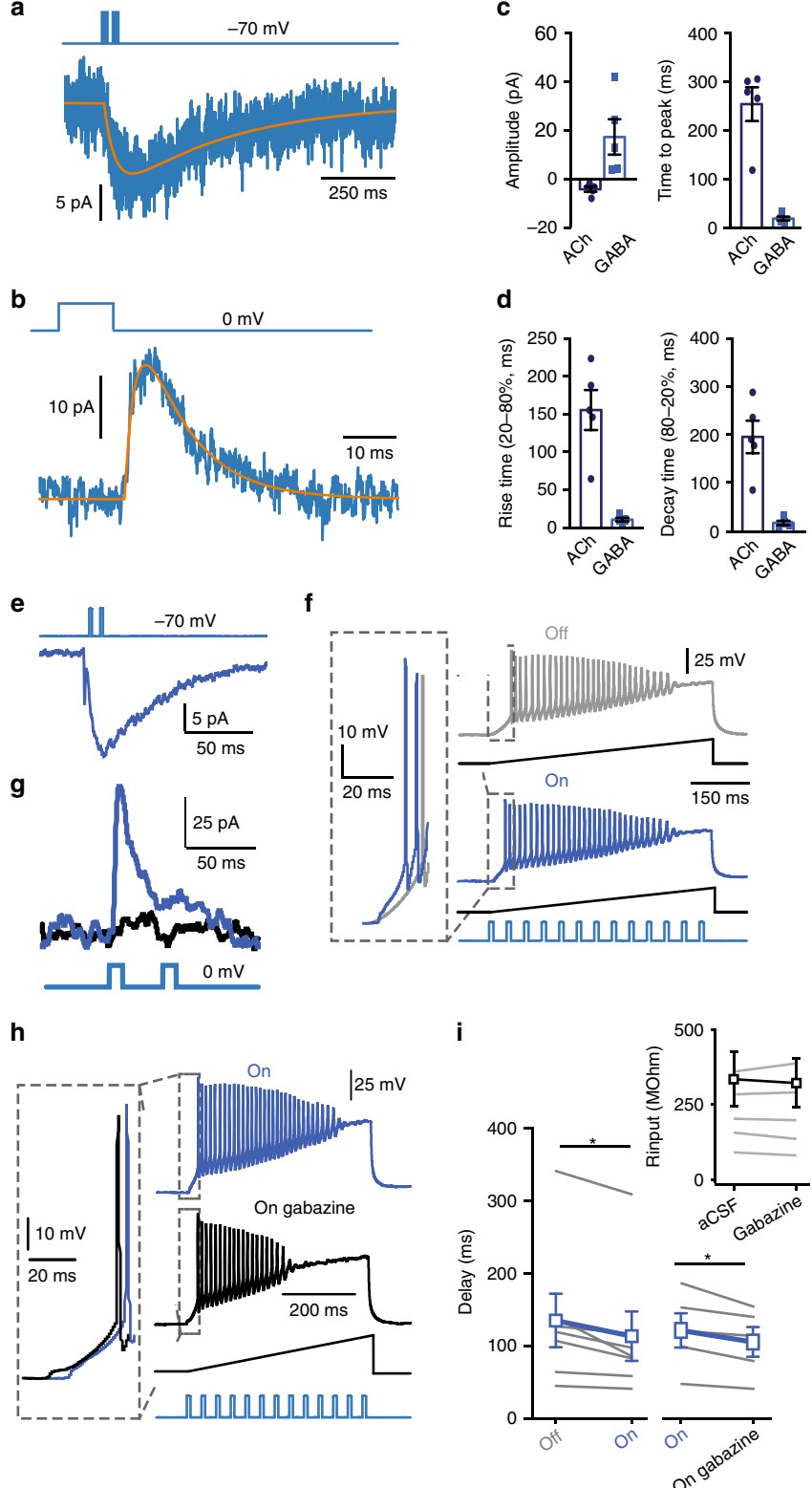

behavioural parameters were observed, including motor behaviour or motivation to respond as quantified by their response latency and latency to collect the reward (Fig. 6f, g; Supplementary Fig. 6), These results show that the activity of BF cholinergic projections to the mPFC and the activity of local ChAT-VIP cells within the mFPC are both involved with proper attention performance.

Interference with the cholinergic system can produce fluctuations in attentive engagement in 5CSRTT[45,46]. Analysis of attention performance in distinct temporal phases of 5CSRTT sessions showed that inhibiting cholinergic BF-to-mPFC projections reduced accuracy of responding only during the first half of the session (early trials 0–50), but not the second half of the session (late trials 51–100) (Fig. 6h). In contrast, inhibiting mPFC

**Fig. 5** Co-transmission of GABA and ACh postpones AP spiking. **a** Postsynaptic nAChR-mediated current (blue) recorded at −70 mV, with fitted trace (orange). **b** Same cell as in (**a**) at 0 mV in the presence of nAChR blockers showing the GABAR-mediated postsynaptic current. **c** Summary plots of amplitude and time to peak of recorded nAChR and GABAR currents. **d** Summary plots of rise and decay kinetics of recorded nAChR and GABAR currents. Summary plots show averages ± SEM. **e** L1 interneuron showing an inward current at −70 mV in response to light-evoked ChR2-mediated activation of ChAT-VIP neurons. **f** Example traces showing AP firing in response to a voltage ramp (ramping current injection 1 pA/ms for 500 ms) in control (grey trace) and with ChR2-mediated ChAT-VIP neuron activation (by 13 blue-light pulses, 10 ms, 25 Hz, blue trace). **g** Light-evoked postsynaptic current response in a L1 interneuron held at 0 mV in the absence (blue trace) or presence of gabazine (black trace). **h** As in (**f**) but either with blue-light stimulation (blue trace) or blue-light stimulation in the presence of gabazine (black trace). **i** Summary plots of the time to first AP in cells without and with blue-light-evoked activation of ChAT-VIP interneurons (Left: aCSF: 136 ± 37.06 ms, aCSF + light: 114.2 ± 34.25, $p = 0.0159$, paired $t$-test, two-tailed, $t = 3.326$, $df = 6$, $n = 7$). Right: summarizing the time to first AP in cells containing both nAChR and GABA$_A$R-mediated postsynaptic currents in absence or presence of gabazine (aCSF + light: 96.07 ± 18.7 ms, gabazine: 83.65 ± 16.28, $p = 0.03$, paired $t$-test, two-tailed, $t = 3.268$, $df = 4$, $n = 5$, mean ± SEM). Inset: Input resistance of whole recordings in absence or presence of gabazine.

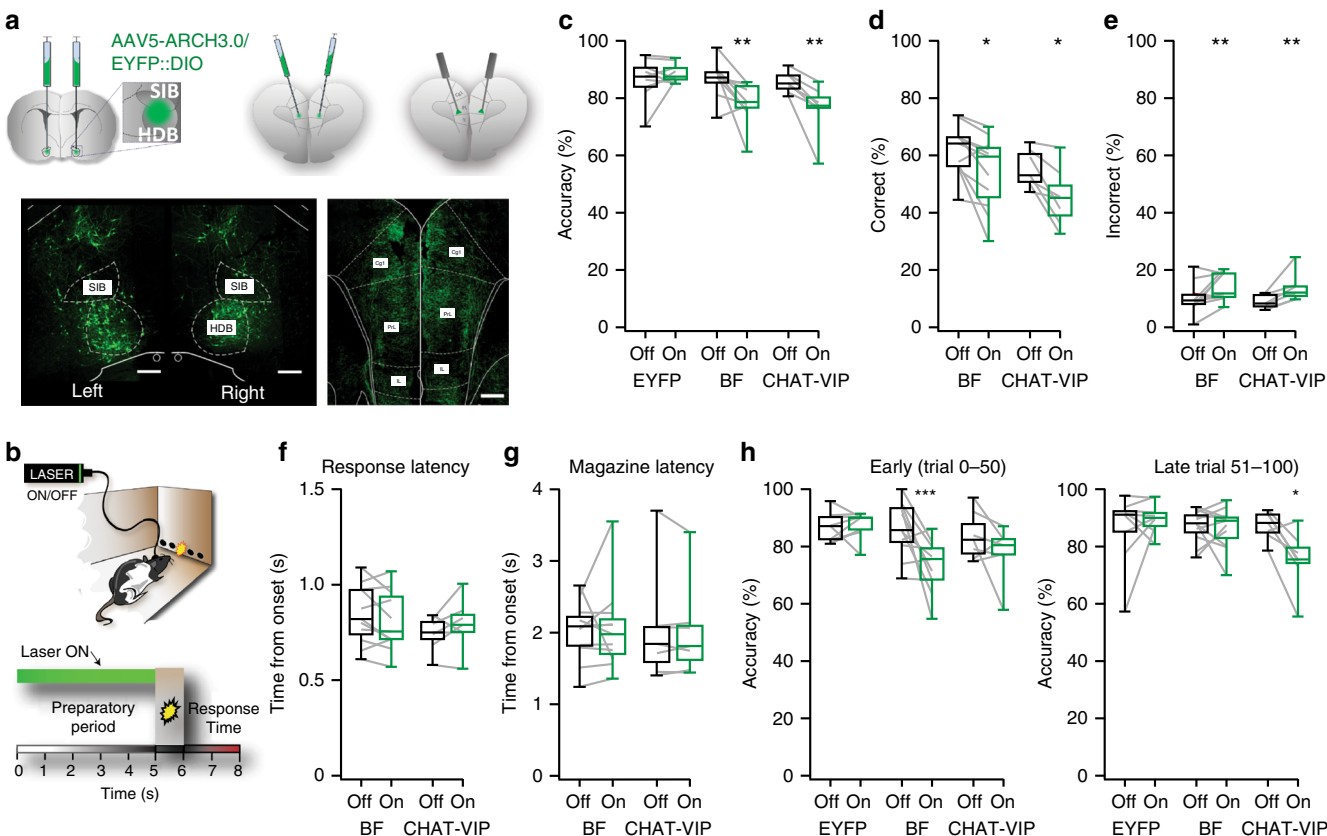

**Fig. 6** ChAT-VIP neurons control attention. **a** Locations of virus injections, in BF (left) or mPFC (middle). Optic fibres were implanted over the PrL mPFC (right). Below: EYFP expression in BF neurons (left image) and EYFP-positive fibres in mPFC (right image). Scale bars 500 μm. **b** Cartoon and schematic representation of a trial. Green bar indicates the period of optogenetic inhibition. Laser-ON and laser-OFF trials were assigned randomly in each session (50 ON, 50 OFF). **c** Accuracy of responding by rats injected either in BF or mPFC (CHAT-VIP), or control littermates that received only AAV5::DIO-EYFP injections [CTRL: $n = 9$; ChAT-VIP: $n = 7$; BF: $n = 11$; two-way ANOVA, effect of interaction light × virus F$_{(2,24)} = 5.920$; $p = 0.0081$; Sidak's correction ChAT-VIP: $p = 0.0102$ ON vs. OFF; Sidak's correction BF: $p = 0.0030$ ON vs. OFF]. Green boxes represent laser-ON trials, black boxes represent laser-OFF trials. Box plot centre line: median; bounds of box: 25th to 75th percentile; whiskers: min-max data bound. **d** Percent of correct responses (two-way ANOVA; effect of interaction light × virus F$_{(2,24)} = 4.088$; $p = 0.0297$; Sidak's correction ChAT-VIP: $p = 0.0214$, BF: $p = 0.0288$ ON vs. OFF). **e** Percent of incorrect responses (two-way ANOVA; effect of interaction light × virus F$_{(2,24)} = 3.835$; $p = 0.0359$; Sidak's correction ChAT-VIP: $p = 0.0314$, BF: $p = 0.0254$ ON vs. OFF). **f** Time to respond to light cues (latency correct latency ChAT-VIP: $t = 1.389$; $p = 0.2141$; BF: $t = 1.576$; $p = 0.142$. Incorrect latency ChAT-VIP: $t = 1.173$; $p = 0.2851$; BF: Wilcoxon matched-pairs signed rank test; $p = 0.6523$, ON vs. OFF). Grey shaded area represents the duration of the stimulus light presentation. **g** Time to collect reward at the magazine. Wilcoxon matched-pairs signed rank test; ChAT-VIP: $p = 0.9063$; BF: $p = 0.4785$). Grey shaded area represents the duration of the stimulus light presentation. **h** Accuracy of responding during early trials and late trials [BF projections inhibition in mPFC: two-way ANOVA effect of interaction light × virus: F$_{(2,24)} = 3.744$; $p = 0.0385$; Sidak's correction BF: $p = 0.0022$ ON vs. OFF], [ChAT-VIP cell inhibition in mPFC: two-way ANOVA effect of interaction light × virus F$_{(2,24)} = 3.744$; $p = 0.0161$ Sidak's correction ChAT-VIP: $p = 0.0125$ ON vs. OFF]. Green bars represent trials with laser-ON. Black bars represent laser-OFF trials. Values are expressed in percent as mean ± SEM. Asterisks: $1 = p < 0.05$, $2 = p < 0.01$, $3 = p < 0.001$.

ChAT-VIP interneurons significantly reduced attention performance in the second half of the session (Fig. 6h). These results indicate that BF ChAT neurons and mPFC ChAT-VIP interneurons affect attention performance distinctly: BF cholinergic neurons support early phases of attention performance, while activity of mPFC ChAT-VIP interneurons is required to sustain attention during the late phase of the session.

## Discussion

In this study, we asked how cortical ChAT-VIP interneurons affect local circuitry in the mPFC, whether they function similar to other cortical VIP cells and whether they are involved in attention behaviour. We found that ChAT-VIP interneurons release ACh locally in both mouse and rat mPFC and directly excite interneurons and pyramidal neurons in different layers via fast synaptic transmission. In contrast to regular VIP interneurons, this ChAT-expressing subtype of VIP interneurons does not inhibit neighbouring fast-spiking and low-threshold spiking interneurons. Our experiments revealed that activity of ChAT-VIP interneurons contributes to attention behaviour in a distinct manner from activity of basal forebrain ACh inputs to mPFC: ChAT-VIP neurons support sustained attention. These findings challenge the classical view that behaviourally relevant cholinergic modulation of neocortical circuits originates solely from BF cholinergic projections in rodent brain[47].

Various reports over the last 30 years identified neocortical ChAT-expressing VIP interneurons and these were suggested as a local source of ACh in the cortex[13–19,25]. Simultaneous recordings from cortical ChAT-VIP interneurons and pyramidal neurons showed an AChR-dependent increase of excitatory inputs received by pyramidal neurons following high frequency stimulation of ChAT-VIP interneurons[19]. However, no evidence was found for direct cholinergic synaptic transmission between ChAT-VIP and other neurons in cortical L2/3[48]. We took a different approach from previous studies by recording from neuron populations that have strong nAChR expression[36,39,49], i.e. L1 interneurons and L6 pyramidal neurons in both mouse and rat mPFC, as well as using ChR2-mediated activation of ChAT-VIP neurons. Our results suggest that ChAT-VIP interneurons form fast cholinergic synapses onto local neurons, since in unitary synaptic recordings the delay between presynaptic AP and postsynaptic response was 1.26 ms, suggesting mono-synaptic connections. Therefore, it is unlikely that ChAT-VIP interneurons triggered poly-synaptic events, exciting terminals of BF neurons and triggering ACh release from these terminals. Fast cholinergic inputs from ChAT-VIP neurons are more abundant in rat mPFC L1 interneurons than in mouse mPFC, in line with a larger percentage of VIP cells expressing ChAT in rat cortex[16]. Von Engelhardt et al.[19] did not observe nAChR currents activated by mouse ChAT-VIP cells in other L3 interneurons, which we did find in rat mPFC. This may be due to species differences or brain region difference in the two studies. Nevertheless, our findings show that in addition to cholinergic fibres from the BF, ChAT-VIP interneurons act as a local source of ACh modulating neuronal activity in mPFC.

Regular cortical VIP interneurons disinhibit local pyramidal neurons by selectively inhibiting somatostatin (SST) and parvalbumin (PV)-expressing interneurons[26,28,30,31,50]. In contrast, we did not find evidence that ChAT-VIP neurons form disinhibitory circuits. Low-threshold spiking and fast-spiking interneurons receive exclusively cholinergic excitatory inputs and no GABAergic inhibitory input from ChAT-VIP interneurons. Prefrontal cortical ChAT-VIP neurons also did not

indirectly disinhibit L2/3 pyramidal neurons through excitation of L1 interneurons[34]. In mouse auditory cortex, fear-induced activation of L1 interneurons by cholinergic inputs from the BF results in feed-forward inhibition of L2/3 FS interneurons and disinhibition of L2/3 pyramidal neurons[34]. ChAT-VIP interneurons in the mPFC might in principle play a similar role in exciting L1 interneurons as BF cholinergic inputs do in mouse auditory cortex. However, in our experiments we did not find evidence that ChR2-mediated activation of ChAT-VIP neurons altered ongoing inhibition and spontaneous inhibitory inputs to L2/3 pyramidal neurons. In contrast, we found that ChAT-VIP interneurons directly targeted a subgroup of L2/3 pyramidal neurons and provided direct excitation to these pyramidal neurons.

Recent anatomical and functional evidence shows that VIP interneurons in rodent brain are morphologically and functionally diverse and that prefrontal cortical VIP cells can directly target pyramidal neurons[51–53]. Both multipolar and bipolar VIP cells form synapses on apical and basal dendrites of pyramidal neurons in superficial and deep layers and VIP neurons directly inhibit pyramidal neuron firing[51–53]. Frontal cortical VIP cells rapidly and directly inhibit pyramidal neurons, while they can also indirectly excite these pyramidal neurons via parallel disinhibition. These findings suggest that not all VIP cell subtypes adhere to targeting only other types of interneurons, and regulating cortical activity through disinhibition only. VIP interneurons represent about 15% of all cortical interneurons in mouse brain, and recent single cell RNA sequencing profiling identified 12 different molecular VIP-positive subtypes, of which 3 types express ChAT[18,25]. Our findings show that ChAT-VIP interneurons project to both interneurons as well as pyramidal neurons and directly excite them, in contrast to most regular VIP interneurons. Thereby, activation of ChAT-VIP interneurons in L2/3 of the mPFC can lead to increased excitability of inhibitory as well as excitatory neurons.

In mouse brain, cholinergic fibres from BF neurons can co-transmit the excitatory neurotransmitter ACh with the inhibitory neurotransmitter GABA in the cortex[41,42,54]. We find here that in rat mPFC, a minority of L1 interneurons (11%) and L6 pyramidal neurons (21%) receive co-transmission of GABA and ACh from the ChAT-VIP interneuron population. Whether both neurotransmitters are actually released from the same ChAT-VIP neuron awaits direct demonstration, but may be likely given simultaneous GAD and ChAT expression by ChAT-VIP neurons. How these two neurotransmitters interact with each other, and what the effect on postsynaptic neurons is, was under debate[42,55]. Nicotinic AChRs show a range of activation kinetics. Heteromeric β2-subunit-containing nAChR currents have relatively slow activation kinetics with 20–80% rise time of 150 ms, while Arroyo et al.[20] showed that homomeric α7-subunit-containing nAChR currents activate rapidly with time constants of 2.6 ms and decay time constants of 4.9 ms in neocortical L1 interneurons[20], comparable to kinetics of synaptic GABAergic currents. This suggests that when the fast α7-subunit-mediated nAChR currents are induced in L1 interneurons by activation of ChAT-VIP cells, the additional GABAR currents that have similar kinetics will shunt the cholinergic depolarization. In case L1 interneurons express only the slower β2-subunit-containing nAChR currents, co-transmission of GABA could augment the excitatory action of ChAT-VIP neurons by rebound excitation. However, depolarizing ramps or near-threshold AP firing probabilities revealed that GABA acted inhibitory in both cases, decreasing spiking probability. Therefore, co-transmission of GABA in addition to ACh postpones AP firing in postsynaptic neurons compared with

synaptic transmission of only ACh, forcing a temporal window of inhibition followed by excitation.

This scheme of postsynaptic GABAR current and AChR current interaction will depend on the physical mode of release, whether these neurotransmitters are released from the same ChAT-VIP cell and the same terminals or not[42,54]. In our experiments using wide-field illumination to activate ChR2 on multiple ChAT-VIP neurons simultaneously, we could not distinguish whether ACh and GABA were released from the same nerve terminals or even from the same ChAT-VIP neuron. It is also not known whether GABA and ACh are packaged in the same vesicles or separately. As such, it is not clear whether co-transmission of GABA and ACh occurs from single ChAT-VIP neurons. However, it is unlikely that ChAT-VIP neurons release only GABA in the mPFC, since we never observed isolated postsynaptic responses mediated only by GABARs, whereas all postsynaptic responses in rat L1 interneurons following ChAT-VIP neuron activation consist of only AChR currents. Thus, regardless of the mode of co-transmitter release, ChAT-VIP activity results in excitation and increased spiking probability throughout the mPFC layers. In addition to depolarization by nAChR activation, calcium influx through these receptors[56] could potentially affect postsynaptic cell function.

Cholinergic signalling in the mPFC controls attention and task-related cue detection[4,11,48,57]. In contrast to the general view that ACh is solely released in the mPFC from cholinergic projections from neurons located in the BF, we present evidence that there is a second source of ACh that supports attentional performance. The different temporal requirements of activity of BF-mPFC projections and ChAT-VIP interneurons in attention suggests that the two sources of cortical ACh interact in shaping cortical network activity during attentional processing. Our findings indicate that activity of cholinergic projections from the BF is required for early phases of attention performance. In contrast, activity of ChAT-VIP interneurons supports later phases of the attention task. Given the sparseness of these neurons, only 15–30% of VIP interneurons express ChAT[16,18], it is surprising that inhibition of this small population in a single brain region has an effect on brain function and behaviour. Even though activation of archaerhodopsin expressed by ChAT-VIP cells or axons may lead to increased activity[58] or suppression of activity, our experiments do show that specific manipulation of these cell populations affect attention. How synaptic connectivity in local PFC circuitry and activity of these neurons actually gives rise to or contributes to attention behaviour in a mechanistic sense is beyond our understanding at this moment.

Recent findings indicate that BF cholinergic neurons are preferentially activated by reward and punishment, rather than attention[59]. Hangya et al.[59] suggested that the cholinergic basal forebrain may provide the cortex with reinforcement signals for fast cortical activation, preparing the cortex to perform a complex cognitive task in the context of reward. Still, rapid transient changes in ACh levels in the mPFC may support cognitive operations[5] and may mediate shifts from a state of monitoring for cues, to generation of a cue-directed response[11,57]. Since we find that activity of ChAT-VIP neurons is required during sustained attention, it remains to be determined whether ACh release from local ChAT-VIP interneurons is responsible for or contributes to the generation of cue-directed responses.

## Methods
**Animals**. All experimental procedures were in accordance with European and Dutch law and approved by the animal ethical care committees of the VU University and VU University Medical Center, Amsterdam. Mice: experiments were done on acute brain tissue of both female and male ChAT-IRES-Cre mice (JAX

laboratory, mouse line B6;129S6-Chattm2(cre)Lowl/J[33]). Average age at time of injection was 9 weeks; average age at time of sacrifice was 16 weeks. Rats: male ChAT-cre rats (kindly provided by the Deisseroth lab[37]) were bred in our facility, individually housed on a reversed 12 h light-dark cycle (lights OFF: 7 a.m.) and were 12–13 weeks old at experiment start. Only when assigned to behavioural experiments, rats were food deprived (start 1 week before operant training, 85–90% of the free-feeding body weight). Water was provided ad libitum.

**Surgical procedures**. All coordinates of injection and fibre placements are from the Rat Brain Atlas (Paxinos and Watson). Viruses AAV5.EF1a.DIO.hChR2.EYFP; AAV5.EF1a.DIO.EYFP and AAV5.EF1a.DIO.eARCH3.0 (titre 4.3–6.0 × 10$^{12}$ ml$^{-1}$) were purchased from UPENN Vector Core (Pennsylvania, USA). Following anaesthesia (isoflurane 2.5%) and stereotaxic frame mounting (Kopf instruments, Tujunga, USA), the scalp skin was retracted and two holes were drilled at the level of either the basal forebrain (BF) or the medial prefrontal cortex (mPFC). Stainless steel micro-needles connected to syringes (Hamilton, USA) were inserted to deliver virus. To optimize rat BF injection location, as we previously did for mouse BF[6], four BF coordinates were used: (a) AP −1.20 mm; ML 2.0 mm; DV −6.8 and 8.9 (1 μl in total) or −7.8 mm (0.5 μl) from skull; (b) AP −0.60 mm; ML 2.0 mm; DV −8.4 mm from skull; (c) AP 0.00 mm; ML 1.6 mm; −8.7 and −8.4 (1 μl in total) or −8.6 mm (0.5 μl) from skull; (d) AP + 0.84 mm; 0.9 mm; DV −7.9 and −8.3 (1 μl in total) or −8.1 mm (0.5 μl) from skull. For behavioural experiments, injection location in BF was used that resulted in highest EYFP expression in BF to mPFC projection fibres (AP 0.00 mm; ML 1.6 mm; DV −8.7 and −8.4 mm from skull). For mPFC injections were done at AP +2.76 mm; ML 1.35 mm; DV −3.86 and −4.06 mm from skull. For the latter group an infusion angle of 10° was employed[60]. In all cases, for behavioural experiments 1 μL virus was injected per hemisphere in two steps of 500 nL, at 6 μL h$^{-1}$ infusion rate.

Mice were two to three months of age at time of surgery and virus injection. Analgesia was established by subcutaneous injection of Carprofen (5 mg/kg) and Buprenorphine (100 μg kg$^{-1}$) followed by general anaesthesia with Isoflurane (1–2%). AAV5 virus (EF1a.DIO.hChR2.EYFP) was injected in both hemispheres (400–500 nL per hemisphere) of the mPFC (coordinates relative to Bregma: AP −0.4, −0.4; ML −1.8 mm; DV −2.4, −2.7) with a Nanoject (Drummond). Mice were sacrificed for experiments at least 3 weeks post-surgery.

Following virus delivery in rat brain for behavioural experiments, two guide screws and two chronic implantable glass fibres (200 μm diameter, 0.20 numerical aperture, ThorLabs, Newton, NJ, USA) mounted in a sleeve (1.25 mm diameter; ThorLabs, Newton, NJ, USA) were placed over the Prelimbic mPFC (200–300 μm on average) under a 10° angle[58]. Finally, a double component dental cement (Pulpdent, Watertown, USA) mixed with black carbon powder (Sigma-Aldrich, USA) was used to secure optic fibres. All surgical manipulations were performed prior to behavioural training and testing.

**Acute brain slice experiments**. Coronal slices of rat or mouse mPFC injected with ARCH3.0 or ChR2 were prepared for electrophysiological recordings. Rats (3–5 months old) were anesthetized (5% isoflurane, i.p. injection of 0.1 ml/g Pentobarbital) and perfused with 35 ml ice-cold N-Methyl-D-glucamin solution containing (in mM): NMDG 93, KCl 2.5, NaH$_2$PO$_4$ 1.2, NaHCO$_3$ 30, HEPES 20, Glucose 25, NAC 12, sodium ascorbate 5, sodium pyruvate 3, MgSO$_4$10, CaCl$_2$ 0.5, at pH 7.4 adjusted with 10 M HCl. Following decapitation, the brain was carefully removed from the skull and incubated for 10 min in ice-cold NMDG solution. Medial PFC brain slices (350 μm thickness) were cut in ice-cold NMDG solution and subsequently incubated for 3 min in 34 °C NMDG solution. Before recordings, slices were incubated at room temperature for at least 1 h in an incubation chamber filled with oxygenated holding solution containing (in mM): NaCl 92, KCl 2.5, NaH$_2$PO$_4$ 1.2, NaHCO$_3$ 30, HEPES 20, Glucose 25, NAC 1, sodium ascorbate 5, sodium pyruvate 3, MgSO$_4$ 0.5, CaCl$_2$ 1. Mouse brains were sliced in an ice-cold sucrose-based solution (in mM: Sucrose 70, NaCl 70, KCl 2.5, NaH$_2$PO$_4$ 1.25, MgSO$_4$ 5, CaCl 1, D-glucose 25, NaHCO$_3$ 25, sodium ascorbate 1, sodium pyruvate 3) and subsequently transferred to aCSF. Standard equipment for whole-cell recordings were used in the following[24]: Borosilicate glass patch-pipettes (3–6 MΩ resulting in access resistances typically between 7 and 12 MΩ), Multiclamp 700B amplifiers (Molecular Devices), and data were collected at 10 kHz sampling and low-pass filtering at 3 kHz (Axon Digidata 1440 A and pClamp 10 software; Molecular Devices).

Recordings from animals injected with ChR2 were made at 32 ± 1 °C in oxygenated aCSF containing in mM: NaCl 125, KCl 3, NaH$_2$PO$_4$ 1.25, MgSO$_4$ 1, CaCl$_2$ 2, NaHCO$_3$ 26, Glucose 10. In all of these recordings in rats, antagonists to block AMPA receptors 6,7-dinitroquinoxaline-2,3-dione (DNQX, 10 μM), NMDA receptors DL-2-Amino-5-phosphonopentanoic acid (DL-AP5, 25 μM) and muscarinic receptors Atropine (400 nM) were bath applied.

For blocking nAChRs the following antagonists were bath applied: DHßE (10 μM) and Methyllycaconitine (MLA, 100 nM). GABAA receptor mediated responses were blocked by bath application of the antagonist gabazine (10 μM). For whole-cell recordings of EYFP-positive ChAT-VIP interneurons and other L2/3 interneurons a potassium-based internal solution was used containing (in mM): K-gluconate 135, NaCl 4, Hepes 10, Mg-ATP 2, Phosphocreatine 10, GTP (sodium salt) 0.3, EGTA 0.2. During recordings, ChAT-VIP interneurons were kept at a

membrane potential of −70 mV. Whole-cell recordings of L1 interneurons and pyramidal neurons were made using a caesium gluconate-based intracellular solution containing in mM: Cs gluconate 120, CsCl 10, NaCl 8, MgATP 2, Phosphocreatine 10, GTP (sodium salt) 0.3, EGTA 0.2, HEPES 10. For paired recordings between ChAT-VIP interneurons and L1 interneurons, potassium-based internal solution was used for both cells. Interneurons and pyramidal neurons were identified by their morphology under IR-DIC, the distance of the soma to the pia and their spiking profile. Membrane potentials were kept at −70 or 0 mV to investigate nAChR or GABAR currents.

Opsins were activated by green (530 nm, eARCH3.0) or blue light (470 nm, ChR2). Light pulses with the specific wavelengths were applied to the slices by using a Fluorescence lamp (X-Cite Series 120q, Lumen Dynamics) or a DC4100 4-channel LED-driver (Thorlabs, Newton, NJ) as light source. During recordings from brain slices from animals injected with eARCH3.0, 20 sweeps, each 10 s apart were applied. One sweep consists of a 1-s long light pulse. The intensity of the light source was adjusted to 1.7, 3, 7 and 17 W.

**Immunohistochemistry.** For processing of brain slices with biocytin-filled neurons (Supplementary Fig. 2) brains from AAV5.EF1a.DIO.EYFP-injected ChAT-cre rats were sectioned in 30-μm-thick slices[22,37]. BF and mPFC slices were stored in PBS overnight and subsequently incubated in citrate buffer pH 6.0 for 3 × 10 min. Thereafter sections were incubated with heated citrated buffer with 0.05% Tween-20 at 90 °C for 15 min, left to cool down, and subsequently, rinsed with 0.05 M TBS. Next, sections were incubated overnight in 0.05 M TBS with 0.5% triton (Tx) containing all five primary antibodies as a cocktail at room temperature. After rinsing slices with TBS (3 × 10 min), sections were incubated for 2 h with secondary antibodies in TBS-Tx. Finally, slices were rinsed in Tris-HCL and mounted on glass slides in 0.2% gelatin, dried, mounted with Mowiol (hecht assistant 1.5 H cover-slips). As controls, adjacent sections were included for all five labels.

ChAT staining (Supplementary Fig. 3) was performed with anti-ChAT raised in goat (1:300, AB144P, Chemicon Millipore, France) and Alexa Fluor-568-conjugated donkey anti-goat (1:400; A11057, Molecular Probe, Fisher Termo Scientific, Waltham, MA). GAD67 staining was performed with primary antibody anti-GAD67 raised in mouse (1:1200, MAB5406 clone 1G10.2, Chemicon Millipore) and visualized using donkey anti-mouse Alexa 546 (1:400, A10036, Molecular probe). VIP staining was performed with rabbit anti-VIP (1:1200, 20077 ImmunoStar, Hudson, WI) and donkey Alexa-anti-rabbit 594 (1:400, A21207 Molecular probe) as secondary antibody. Further, guinea pig-anti-calretinin (1:4000, 214104, Synaptic systems, Goettingen, Germany) together with donkey-anti-guinea pig Alexa 647 (1:400, Jackson 706-605-148).

**Cell counts in basal forebrain.** To quantify potential retrograde labelling by AAV5 from the mPFC to the BF (Supplementary Fig. 4), rats were injected with AAV5-DIO::eYFP either in the mPFC or the BF at the coordinates used for behavioural and physiological experiments. 50 μm slices of the brains were cut using a vibratome (Leica, 1200T, Germany). Slices were stained for eYFP and mounted on glass slides covered by 2% Mowiol, anti-fading mounting agent and cover slip. Images were acquired using a confocal laser scanning microscope (CLSM; Zeiss LSM 510 Meta) with an excitation wavelength of 514 nm (bandpass 530–600 nm). Cell counting was performed using the cell count function of ImageJ.

**Attention behaviour.** After 1 week of recovery from surgery and 1 week of habituation to the reversed light/dark cycle, rats started training in the 5CSRTT in operant cages (Med Associates Inc., St. Albans, VT, USA) (Supplementary Fig. 6). In short, following the initial training phase, progression was based on individual performance of each rat, and was reduced from 16 to 1 s[60]. Criteria to move to a shortened stimulus duration were the percentage of accuracy (>80%) and omitted trials (<20%). When the criterion of 1 s stimulus duration was reached animals were moved to the pretesting phase. In the pretesting phase, a green custom-made LED replaced the normal house-light of the operant cages (<1 mW intensity) to mask reflections by the laser light used for the experiments.

After three consecutive sessions during which rats performed according to criteria with the LED on in the operant cage, three additional baseline sessions were conducted. During these sessions, rats were connected to the patch-cable (Doric Lenses, Quebec city, Canada) used to deliver the light into the brain. In this condition, percentage accuracy was above 80%. However, rats often did not show <20% omissions within sessions. This was most likely due to the fact that the animals were connected to the optic fibre patch-cable and therefore less free to move in combination with the short time window for the animal to respond (i.e. within 2 s after the cue light went off). Therefore, in line with previous work[60], the omission criterion was increased to <40% omissions.

Following acquisition of baseline performance, rats were assigned to the testing phase where the task comprised 100 consecutive trials with a random assignment of laser-ON or laser-OFF trials. For the testing phase, the following parameters were acquired and analysed through a box-computer interface (Med-PC, USA) and custom-written MATLAB scripts (Mathworks): accuracy on responding to cues (ratio between the number of correct responses per session over the sum between correct and incorrect hits, expressed as percentage); absolute and percentage of

correct, incorrect responses and errors of omission; correct or incorrect response latency; latency to collect reward; number of premature and perseverative responses. Percent of correct, incorrect and omissions were calculated based on the number of started trials[61] to allow a more sensitive evaluation of the parameters.

**Optical inhibition during behaviour.** To light-activate the opsins in vivo, we used a diode-pumped laser (532 nm, Shanghai Laser & Optics Century Co, China) directly connected to the rat optic glass fibre implant. Light was delivered at 7–8 mW from the fibre tip for experiments carried out with eARCH3.0. These stimulation regimens are able to produce a theoretical irradiance which ranges between 7.59 and 8.68 mW mm$^{-2}$ (http://web.stanford.edu/group/dlab/cgi-bin/graph/chart.php). Light was delivered according to scheduled epochs by a stimulator (Master 9, AMPI Jerusalem, Israel) connected to the computer interface, which semi-randomly assigned the different trials to laser-OFF or laser-ON conditions (50% of each). In the laser-ON condition, light was delivered during the whole preparatory period (5 s) that precedes stimulus presentation. Optical inhibition sessions were repeated two times per rat with a baseline session in between to control for potential carry-over effects.

Moreover, reported data for the majority of rats refer to the first two optical inhibition sessions after establishment of stable baseline performance. Power analysis based on the effect size determined the minimal sample size to detect a statistical significance (7 or more) with a power of $\beta = 0.9$.

**Histological verification.** After behavioural testing, brains were checked for fibre placement and viral expression. For this, rats were anesthetized with isoflurane and a mix of ketamine (200 mg kg$^{-1}$ i.p.) and dormitol (100 mg kg$^{-1}$ i.p.) and then transcardially perfused (50–100 mL NaCl and 200–400 mL PFA 4%). Brains were removed and maintained in 4% PFA for at least 24 h. After that, brains were sliced with a vibratome (Leica Biosystem, Germany) into 50–100 μm coronal sections and mPFC slices were mounted on glass slides covered by 2% Mowiol, anti-fading agent and cover slipped. Images were taken with a CLSM (LSM 510 Meta; Zeiss, Germany) with excitation wavelength of 514 nm bandpass filtered between 530 and 600 nm, and further analysed using ImageJ (NIH, USA).

**Quantification and statisitical analysis.** To evaluate behavioural performance between the ARCH3.0 groups and EYFP control group, two-way ANOVAs for repeated measures were performed. Corrected values for multiple comparison with Sidak's test were used when the interaction between light and virus was significant. In all cases, the ANOVAs were preceded by the Kolmogorov–Smirnov (KS) test for normal distribution. In cases when the KS p-value was >0.05, factorial analysis was performed on the raw data per parameter. In other cases, raw data were first transformed with square-root or arcsin transformation. Analysis of other parameters were performed with student's t test, Wilcoxon test and always preceded by KS test to check for normal distribution of the sample. Data were analysed by MATLAB 2016a (Mathworks), Microsoft Excel (Office) and graphs were plotted by GraphPad Prism. In all cases the significance level was $p < 0.05$.

To statistically evaluate the results between nAChR blockers and aCSF conditions in acute slice experiments, two-tailed paired Student's t-test was employed. To evaluate differences with GABAR blockers two-way ANOVA for repeated measures was used. If the amplitude of the response was lower than four times the standard deviation of the baseline preceding the stimulation, the event was counted as a failure. The average time for onset delay from AP peak was calculated by fitting a lognormal function to the histogram of onset delays. To quantify the spike delay time and probability two-tailed paired student's t-test was used. Significance level was set to $p < 0.05$.

**Reporting summary.** Further information on research design is available in the Nature Research Reporting Summary linked to this article.

## Data availability
Further information and requests for data, resources and reagents should be directed to corresponding author H.D.M. at h.d.mansvelder@vu.nl.

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

## Acknowledgements
We thank J.N. Vaiña, MSc, for excellent technical assistance, and Dr. K. Deisseroth for the generous sharing of tools and ChAT-cre rats. H.D.M. received funding for this work from the Netherlands Organization for Scientific Research (NWO; VICI grant 865.13.002), ERC StG "BrainSignals," EU H2020 Framework Programme (Grant Agreement H2020 HBP 720270), and European Union's Horizon 2020 Framework Programme for Research and Innovation under the Specific Grant Agreement No. 785907 (Human Brain Project SGA2) and EraNet Neuron/NWO. C.P.J.d.K. received funding for this work from the Netherlands Organization for Scientific Research (NWO ALW #822.02.013).

## Author contributions
H.D.M., T.P., A.L. and J.O. designed the study. A.L. and O.M.N. performed behaviour experiments. A.L., J.O. and O.M.N. performed surgeries, perfusions and anatomy experiments. S.D.K., H.T. and B.B. assisted in the training, behaviour and anatomy experiments. R.D.H. and C.P.J.d.K. provided analysis tools and MATLAB scripts. A.L., H.D.M. and T.P. analysed the behavioural data. J.O., T.H., C.K., N.A.G., A.A.G. and A.J.K. performed ex vivo electrophysiology experiments. J.O. and H.D.M. designed and analysed the electrophysiological data. T.K., A.J.J., N.A.G., W.V.D.B. and C.P.J.d.K. performed biocytin- and immunostaining experiments. N.A.G. analysed the scRNAseq data from the Allen Institute database. J.O., A.L., H.D.M. and T.P. wrote the paper with input from all other authors.

## Competing interests
The authors declare no competing interests.
