## [Peer Review File · Nature Communications]

Reviewers' Comments:

Reviewer #1:

Remarks to the Author:

This interesting manuscript by Obermayer et al seeks to understand the cellular/circuit level functions for ChAT-VIP interneurons in the neocortex. While basal forebrain cholinergic modulation of cortical circuit function has been well studied, contributions from neocortical ChAT neurons are not well understood. Using now standard while rigorous electrophysiological and optogenetic approaches in acute slice preparations, the authors investigated the functional connectivity of mPFC L2/3 ChAT-VIP interneurons: 1) with L1 interneurons, L6 pyramidal neurons; 2) whether ChAT-VIP interneurons contribute to L2/3 pyramidal neuron disinhibition; 3) circuit level functions for ACh vs ACh+GABA co-release on L1 interneuron AP initiation timing. The authors compared results from both ChAT-Cre mouse and rat models, and uncovered some interesting differences. Lastly, using in vivo optogenetic silencing, they performed 5 choice serial reaction time task in rats to show functional significance for cortical ChAT-VIP neurons in sustained attention. There is a nice combinatorial experimental approach here to tackle an interesting problem, showing that a different neuronal ensemble source for ACh is potentially important for cortical circuit function. Conceptually, the observations here have some parallels to cholinergic excitation and modulation of local inhibitory interneurons in the striatum. While the results are generally interesting, there are several issues needing to be addressed by the authors.

1) The authors' presentation is currently a collection of several smaller questions focused on cortical ChAT-VIP interneurons. Many "unknowns" were stated in the Introduction. The significance and progress of solving each smaller question appeared incremental in overall conceptual advance... Although the behavioral experiments at the end tried to tie the in vitro and in vivo results together, it wasn't clear how circuit level modulations contributed to attention. Following optogenetic silencing, were there activity-dependent gene expression/marker changes in L1 interneurons, L2/3 or L6 pyramidal neurons? A complementary experiment to optogenetically activate ChAT-VIP interneurons in vivo was not performed. From an impact standpoint, I do wonder if the overall question is 1) whether other sources of ACh (besides basal forebrain) can significantly modulate cortical circuit function during attention tasks; 2) hypothesizing cortical ChAT-VIP interneurons may play a role; 3) in vivo behavioral studies; 4) in vitro physiological experiments to dissect ChAT-VIP function. In my opinion, the manuscript's current presentation lessened its impact.

2) This study relied on ChAT-Cre driver lines in mice and rats. The differences in results between mice and rats are potentially interesting, but they may also be due to specificity differences between mouse and rat ChAT-Cre driver lines in targeting cortical ChAT neurons. Rigorous antibody labeling/anatomical studies need to be performed using Cre-dependent viral reporters, to ensure differences observed between mice and rats = "biology", rather than specificity differences between Cre drivers.

3) page 5, line 105: "EYFP-positive neurons showed similar morphology, ChAT, VIP, CR, GAD expression patterns..." It is rather important for the functional studies throughout that ChAT-VIP neurons targeted are not a diverse cellular population. However, data presented in Supplementary Fig 1 was not convincing. Quality of GAD67 staining in Fig S1A was surprisingly poor and needs to be repeated. In Fig S1B, the single channel EYFP data needs to be shown, and in the 3 color composite image, there appeared to be two EYFP+ neurons that co-labeled with CR but not VIP, while the authors pointed to one EYFP+ neuron co-expressing CR and VIP. I would argue the data suggested that EYFP-positive neurons had different (not similar) CR and GAD expression patterns. If ChAT+/VIP-negative cortical interneurons were targeted, but unaccounted for in the functional and behavioral analyses, that represents a significant problem for the manuscript.

Other points:

- 1) Fig 1F, which inhibitor was used to block the slow component current? This should be clearly described.
- 2) For Supplementary Fig 2A and 2B, AAV vectors will not efficiently travel retrogradely to the basal forebrain, thus AAVretro vectors are needed for validating mPFC injections.
- 3) In Fig 2D and 2F, how were FS and LTS neurons identified experimentally? If fluorescent transgenic reporter lines were used, they need to be stated.
- 4) In describing results from Fig 2A, were the recordings made from rat or mouse slice preparations?
- 5) In Fig 3 results section subheading, "Layer 6 pyramidal neurons receive...", the word "neurons" is missing.
- 6) For Fig 4, the fast MLA-sensitive nAChR current data, was it "data not shown"? It needs to be included.
- 7) In Fig 4F, nAChR inhibitor data needs to be included.
- 8) For the ACh and GABA co-transmission experiments, the authors need to conclusively demonstrate that ACh and GABA are being co-released from the same neuron.

Reviewer #2:

Remarks to the Author:

The paper by Obermayer et al. characterizes cholinergic synaptic transmission between ChAT-VIP positive interneurons in the neocortex and different types of target cells. To address this question, the authors use paired recordings, optogenetics, and behavioral analysis. The main findings are:

- ChAT-VIP interneurons directly excite GABAergic interneurons in layer 1.
- Synaptic transmission is mediated by nicotinic ACh receptors.
- In a subset of recordings, cholinergic transmission is coupled to GABAergic transmission, suggesting co-release of ACh and GABA from the same neurons.
- Inhibition of ChAT-VIP interneurons impairs performance in a 5 choice serial reaction time task in the second phase of the task, whereas inhibiting basal forebrain neurons affects performance in the first phase.

Based on these results, the authors conclude that ChAT interneurons directly excite other interneurons and pyramidal cells via fast cholinergic synaptic transmission. Overall, this is a quite nice paper. The results are interesting, because they represent the first demonstration of fast cholinergic synaptic transmission in brain circuits. Furthermore, the experiments are technically well done, and the manuscript is generally well written. However, I have two major points that need to be addressed in a revision process. First, it remains unclear whether the cholinergic synaptic currents result from vesicular ACh release. Second, the current amplitude is very small, often below 10 pA for unitary synaptic connections. Thus, despite the behavioral data, the physiological significance of such a small current remains somewhat unclear. However, these two points can be addressed in a revision, and do not diminish the potential importance of the findings.

Major points:

1. If nicotinic responses are mediated by quantal release, they should show fluctuating amplitude and show failures of synaptic transmission. This should be documented in the paper.

2. Relatedly, what is the evidence that ChAT-positive interneurons express vesicular ACh transporters? At the very least, this point should be addressed in the discussion.
3. If the cholinergic responses are generated by quantal release of ACh, nicotinic miniature EPSPs should be detectable in layer 1 GABAergic interneurons. The authors should be able to test this.
4. The conclusions largely stand and fall with the paired recordings (the optogenetics is nice, but obviously less compelling). However, the number of experiments is very small. I realize these are difficult experiments, but I feel an $n = 3$ is not enough for a Nature Communications paper.
5. The huge variability in kinetics of the cholinergic EPSCs needs to be better addressed. The authors apparently think that this variability is generated by different subunits of nACh receptors, but this should be tested directly, by applying blockers individually rather than in a cocktail.
6. The authors focus on the excitatory effect of cholinergic synaptic transmission in postsynaptic target cells. Depending on receptor subunit composition, Ca^{2+} inflow through the postsynaptic receptors may be also important. At the very least, this point needs to be addressed in the discussion.
7. The behavioral experiments are potentially interesting, but preliminary. At the very least, the presentation of the data needs to be improved.

Minor points:

Line 114: "about 2 milliseconds" - the exact latency (mean + - SEM) should be specified.

Line 179: "Layer 6 pyramidal" should read "Layer 6 pyramidal cells".

Line 188: It should be possible to provide direct measurements under current clamp conditions, rather than referring back to previous publications.

Line 218: How many cholinergic inputs were stimulated in these experiments?

Line 226: "not shown" - such kind of statements should be replaced by supplementary data.

Line 244: Where is PrL introduced?

Line 349: No evidence for a reliance on beta 2 and alpha seven subunits, respectively, is provided.

Line 351: These time constants are apparently coming from the Hestrin paper, but this should be stated more clearly.

Line 365: "are release from the same" should read "are released from the same".

Line 373: "eighty to ninety percent" - better quantification is needed.

Line 490: Journal is missing from reference.

Line 506: References should be updated. For example, the Tasic et al. paper came out a while ago.

Line 572: Several references are incomplete.

Line 606 "from the same neurons" should be "from the same pair of neurons".

Line 610: "scale bar 200 μm " - figure shows scale bar with 50 μm label. Which information is correct?

Line 623: " μm " should read " μM ".

Line 634: The title of the figure is misleading. Isn't the main point that there is direct cholinergic innervation?

Line 696: "As in showing" - not understandable, revise.

Line 770: Why is the total number of animals given for rats, but not for mice?

Line 819: What is meant by "1 M" ?

Line 824: Additional details about the recording configurations have to be provided. For example, what was the series / access resistance of the recordings? Furthermore, the exact recording temperature, including range, should be specified.

Line 827: "receptors" should be probably "NMDA receptors".

Line 834: "K2Phos" should probably read "K2Phosphocreatine". Furthermore, which GTP salt was used? It should be possible to check the methods section more carefully before submission.

Line 865: Anti-GAD67 - species should be specified (apparently mouse).

Figures: Traces are too thick. This makes it difficult to judge the quality of the recordings.

Figure legends: Are traces shown single traces or averages?

Figure 1E: Where is the axon in this and other cells?

Figure 1I: Without being pedantic, but Y axis labels are shifted.

Figure 2A, pie chart: With 18 cells, the proportion of cells with connections can be either 17%, or 11%, but not 14%. Again, it should be possible to sort of such errors before submission.

Figure 2E, G: Double parentheses should be removed from the y-axis labeling.

Figure 5: Data should be probably presented as whisker plots, as in all previous figures. Furthermore, it would be useful to see data from individual experiments in these graphs.

Legend Figure 5D, E: The authors may want to mention the omissions to explain why correct and incorrect trials don't sum up to hundred percent.

Methods: The quantitative criteria defining the presence or absence of a cholinergic synaptic connection should be better defined. Clearly, the authors cannot exclude the presence of additional

connections with peak amplitudes below a couple of picoampere.

Supplemental figure 1: Both false positives and false negatives should be shown.

Supplemental figure 2: All abbreviations should be defined in the legend.

Supplemental figure 3a: Figure is cut.

Legend supplemental figure 4C: "msec" should be "ms", here and elsewhere in the paper.

Legend supplemental figure 4E, F: These circular definitions are not understandable.

Reviewer #3:

Remarks to the Author:

This manuscript by Obermayer and colleagues studies the local circuit connectivity of a subset of prefrontal VIP neurons expressing ChAT using patch-clamp recordings in slices combined to optogenetic stimulation. It also analyzes the behavioral consequences of selectively inhibiting firing in this population using an optogenetic approach in adult rats performing a 5 choice serial reaction time task. The study includes both mouse and rat experiments (all adults). The major findings of the paper are: (1) that this subpopulation of VIP cells is not involved in canonical "disinhibitory circuits" but rather broadly excites local interneurons (mainly Layer 1) and pyramidal neurons (mainly Layer 6) through direct cholinergic transmission; (2) a fraction of these cells in rats also releases GABA onto L1 interneurons & L6 pyramids; (3) that inhibiting these neurons selectively significantly decreases performance in the later phases of the task, as compared to inhibiting cholinergic inputs from the basal forebrain which have an earlier impact.

This is a potentially interesting finding. The fact that manipulating such a small population of cells (30% of VIP cells) has an impact on behavior (even if quite subtle) is very interesting. It probably reflects their strong connectivity in layer 1 and 6. It is also interesting to see that VIP cells do not only operate in disinhibitory circuits. The paper is well written and the electrophysiological experiments are elegant and well presented. That said, I feel that the title of the paper is little overselling the results. I would not claim that ChAT-VIP interneurons control attention but rather that they "are involved in attention" (the performance decrease is quite mild). I also think that a finer dissection of the specific circuits involving these cells would help understand how their strong network impact may operate. Also, the mix between rat and mouse experiments is sometimes distracting, even more since there seems to be some differences between the two species. In fact, most experiments were performed in rats, except from the paired recordings. In addition, some points require clarification or additional analysis.

More specific comments below:

Postsynaptic partners of the VIP-ChAT neurons: the authors demonstrate the existence of short latency inward currents induced by ChAT-VIP cells onto a large majority of layer 1 interneurons, with all cells responding in rats, in 15% of layer 2/3 pyramids and 71% of rat L6 pyramidal neurons (how about mice?). This is a remarkable connectivity pattern. It would have been interesting to see whether the axonal morphology of VIP-ChAT cells matched this preferential connectivity. Unfortunately, the authors only provide one example of morphological reconstruction, where only the dendrites are recovered.

Also, it is said that 67% of L1 interneurons receive a time-locked inward current in response to the light stimulation but it would be important to specify the criteria for inclusion. What are the criteria to

identify a significant response? Is there for example any threshold-based on response onset delay? Response amplitude? How-time-locked are the responses if the delays can span between 2 and 10ms? The amplitude evoked onto L6 pyramids is very small but the authors state that it "resulted in a significant depolarization of the membrane potential": could this be quantified? How significant is it?

Absence of disinhibition of L2/3 neurons by ChAT-VIP interneurons: the authors measured the changes in sIPSCs frequency received by L2/3 pyramids during light stimulation to support the absence of disinhibition. Is this done in mice and rats? The quantification presented in Fig. 2B is not really convincing since it would be more informative to estimate the relative change in IPSCs frequency per cell (instead of the absolute values). Also, histograms plotting the inter-event intervals as a function of time centered on light stimulation would be more illustrative.

Rat vs mouse: since the authors found differences between species (like no co-release of Ach and GABA in mice), I think it is important to clearly state in the text when experiments were only performed in one species.

Lines 210-211: Hyperpolarizing GABAergic inputs can give rise to rebound excitation also through the activation of the h current (not only deinactivation).

Line 226: Gabazine did not alter excitability in L1 neurons. This is quite surprising given the important contribution of GABAA-Rs to the input resistance. This should be shown and quantified.

Line 238: a few lines describing the task would help the reader.

Line 258: local ChAT-VIP cells are not "required", they may be "involved".

Figure 5: I would avoid using bar graphs (starting from 40% accuracy) but instead provide box plots displaying the variability of the responses. Have the authors tested whether the difference between early and late trial phases was significant for ChAT-VIP stimulation experiments? (it does not look like it when inspecting Fig. 5H). Also, I would like to see the EYFP labeling in the two conditions rather than a cartoon (both low magnification and high magnification displays).

Supplemental Figure 1: Please provide better quality photomicrographs of GAD67 staining as well as lower magnification pictures.

Reviewer #1 (Remarks to the Author):

The authors have improved the manuscript through revision. While several of my technical concerns from initial review have been addressed, the authors' comments to the main issue I had regarding the manuscript, namely it reads as a collection of smaller questions addressing several "unknowns", are unconvincing. What's the main message?

Reply: To make the focus and the main message of the study stand out better, we have adapted the text of the abstract and final paragraph of the introduction.

Reviewer #2 (Remarks to the Author):

1. I continue to think that it would be nice to see examples miniature EPSPs or EPSCs (see my previous major point 3). Even if the cellular sources cannot be identified, this might corroborate a quantal release mechanism. Furthermore, such measurements might provide information about whether ACh is coreleased with GABA, as asked by the other reviewers.

Reply: To solve the reviewer's point, we have done additional experiments, and recorded acetylcholinergic miniature Excitatory PostSynaptic Currents (mEPSCs, see Figure 1A below), as requested by the reviewer. Miniature EPSCs were recorded in mPFC layer 1 interneurons in acute brain slices of adult wildtype BL6 mice in the presence of blockers of voltage-gated sodium channels, AMPA-Rs, NMDA-Rs and GABA-Rs. Fast mEPSCs occurred in all recordings albeit at low frequency and were completely blocked by nicotinic ACh receptor antagonists (Figure 1B). These experiments show that mPFC layer 1 interneurons receive fast cholinergic

mEPSCs, and they confirm our original conclusion on synaptic release of acetylcholine.

Figure 1: Layer 1 interneurons receive cholinergic miniature Excitatory PostSynaptic Currents (mEPSCs).

A: Example traces of mEPSCs (grey) and average current trace (black). (bath solution contained 1 μ M TTX, 10 μ M DNQX, 25 μ M AP5, 10 μ M gabazine, 4 mM calcium. The intracellular solution contained CsCl).

B: The number of mEPSCs observed in two minute time bins is rapidly reduced to zero when antagonists (MLA and DH β E) of nicotinic Acetylcholine receptors (nAChRs) are applied (n=5 of 5 recorded neurons).

C: left: Amplitude distribution of all recorded mEPSCs (n=5 neurons). Despite the modest number of mEPSCs (n=67 events) recorded in total, the histogram might suggest multiple amplitude peaks (quanta). Right: average mEPSC amplitude.

Since these data do not allow any conclusions on the cellular source of the mEPSCs or co-release of ACh and GABA from ChAT-VIP neurons, the figure does not fit the present manuscript.

2. I continue to think that it is essential to specify the precise recording temperature (mean value, range), rather than specifying the approximate value.

Reply: We agree with the reviewer and have now added the temperature range in the methods section.

Apart from these points, this is a nice paper that definitely should be published in Nature Communications.

Reviewer #3 (Remarks to the Author):

This revised version of the manuscript partly (and sometimes superficially) addressed the points I had raised. The finding that mPFC ChAT-VIP interneurons are involved at different phases of attention performance than BF ChAT neurons is interesting. Their specific connectivity patterns as well. This paper is not attempting to link both. How the local cholinergic connectivity scheme relates to attention is not even discussed. For example, the late involvement of the mPFC ChAT-VIP neurons suggests that their inputs or their synapses may display some form of plasticity during tasks requiring long attention. This would have been nice to study, or at least to discuss.

Reply: We agree with the reviewer and have added a sentence to the discussion to highlight this point.

Points that were only partly addressed:

-axonal morphology of VIP-ChAT cells was only superficially addressed with an example picture of EYFP but no cell axonal morphology reconstruction.

Reply: As requested, we have now added images of biocytin-labeled ChAT-VIP neurons with axons from mouse mPFC to Supplementary Figure 1. From these images it is clearly visible that axons of ChAT-VIP neurons abundantly spread to layer 1 and to deep cortical layers 5 and 6, as we show in Figure 4A and as was beautifully described and extensively quantified by Prönneke et al., Cereb Cortex 2015. In addition, the images exemplify the reported bipolar and multipolar soma-dendritic morphologies of ChAT-VIP neurons as reported extensively in the literature (Eckenstein and Thoenen, 1983, 1984; Von Engelhardt et al., 2007), and as shown in Figure 2A of the present manuscript.

- the criteria to identify a significant response are not explained in detail; the authors mainly chose to quantify average delays.

Reply: We thank the reviewer for pointing this unclarity out. The criteria are specified in the Methods section in the paragraph on Quantification and Statistical Analysis: "If the

amplitude of the response was lower than 4 times the standard deviation of the baseline preceding the stimulation', the event was counted as a failure."

In addition to quantifying average delays, we show in Figure 1E and 1H histograms of individual synaptic events, revealing the full distribution of synaptic delays.

- Amplitude of depolarization evoked onto L6 pyramids: I understand that neurons displaying a high resistance can amplify their membrane depolarization in response to small input currents. I was just asking for some quantification of the recorded EPSP.

Reply: Apologies for the misunderstanding. The quantification of the recorded EPSPs are provided in Figure 4D, left panel.

Additional minor points:

VIP-ChAT: from supplementary Figure 1C it does not seem that all EYFP expressing neurons also expressed VIP. Therefore, I feel that the term VIP-ChAT is somehow misleading; this simplified terminology should be better stated and acknowledged.

Reply: ChAT-VIP cells can express VIP and ChAT at different levels. Antibodies can fail to detect low abundant protein, but absence of antibody staining does not necessarily mean complete absence of VIP protein. To illustrate this point, Prönneke et al., 2015 observed in Vip-ires-cre mice crossed with homozygous Ai9 mice introducing a cre-dependent TdTomato reporter, that the VIP antibody staining did not 100% overlap with TdTomato, so some VIP cells (called by the TdTomato expression) were not VIP positive when judged from the antibody staining (see Figure 2A and 2A' in Prönneke et al., 2015). We have added this information to the legend of supplementary Figure 1C.

Lines 129-132 : this sentence reads quite awkwardly, I think it should be placed elsewhere, maybe in the introduction or at the beginning of the results.

Reply: The sentence has now been altered.

Line 152: "than" instead of "then"

Reply: Corrected

Reviewers' Comments:

Reviewer #1:

Remarks to the Author:

The authors have improved the manuscript through revision. While several of my technical concerns from initial review have been addressed, the authors' comments to the main issue I had regarding the manuscript, namely it reads as a collection of smaller questions addressing several "unknowns", are unconvincing. What's the main message? In the present format, I see this paper as more specialized (J Neurosci perhaps), and not at a comparable level to other Nature Communications publications.

Reviewer #2:

Remarks to the Author:

The authors have carefully addressed my comments. However, a couple of minor points remain.

1. I continue to think that it would be nice to see examples miniature EPSPs or EPSCs (see my previous major point 3). Even if the cellular sources cannot be identified, this might corroborate a quantal release mechanism. Furthermore, such measurements might provide information about whether ACh is coreleased with GABA, as asked by the other reviewers.

2. I continue to think that it is essential to specify the precise recording temperature (mean value, range), rather than specifying the approximate value.

Apart from these points, this is a nice paper that definitely should be published in Nature Communications.

Reviewer #3:

Remarks to the Author:

This revised version of the manuscript partly (and sometimes superficially) addressed the points I had raised. The finding that mPFC ChAT-VIP interneurons are involved at different phases of attention performance than BF ChAT neurons is interesting. Their specific connectivity patterns as well. This paper is not attempting to link both. How the local cholinergic connectivity scheme relates to attention is not even discussed. For example, the late involvement of the mPFC ChAT-VIP neurons suggests that their inputs or their synapses may display some form of plasticity during tasks requiring long attention. This would have been nice to study, or at least to discuss.

Points that were only partly addressed:

- axonal morphology of VIP-ChAT cells was only superficially addressed with an example picture of EYFP but no cell axonal morphology reconstruction.
- the criteria to identify a significant response are not explained in detail; the authors mainly chose to quantify average delays.
- Amplitude of depolarization evoked onto L6 pyramids: I understand that neurons displaying a high resistance can amplify their membrane depolarization in response to small input currents. I was just asking for some quantification of the recorded EPSP.

Additional minor points:

VIP-ChAT: from supplementary Figure 1C it does not seem that all EYFP expressing neurons also

expressed VIP. Therefore, I feel that the term VIP-ChAT is somehow misleading; this simplified terminology should be better stated and acknowledged.

Lines 129-132 : this sentence reads quite awkwardly, I think it should be placed elsewhere, maybe in the introduction or at the beginning of the results.

Line 152: "than" instead of "then"

Reviewer #1 (Remarks to the Author):

The authors have improved the manuscript through revision. While several of my technical concerns from initial review have been addressed, the authors' comments to the main issue I had regarding the manuscript, namely it reads as a collection of smaller questions addressing several "unknowns", are unconvincing. What's the main message?

Reply: To make the focus and the main message of the study stand out better, we have adapted the text of the abstract and final paragraph of the introduction.

Reviewer #2 (Remarks to the Author):

1. I continue to think that it would be nice to see examples miniature EPSPs or EPSCs (see my previous major point 3). Even if the cellular sources cannot be identified, this might corroborate a quantal release mechanism. Furthermore, such measurements might provide information about whether ACh is coreleased with GABA, as asked by the other reviewers.

Reply: To solve the reviewer's point, we have done additional experiments, and recorded acetylcholinergic miniature Excitatory PostSynaptic Currents (mEPSCs, see Figure 1A below), as requested by the reviewer. Miniature EPSCs were recorded in mPFC layer 1 interneurons in acute brain slices of adult wildtype BL6 mice in the presence of blockers of voltage-gated sodium channels, AMPA-Rs, NMDA-Rs and GABA-Rs. Fast mEPSCs occurred in all recordings albeit at low frequency and were completely blocked by nicotinic ACh receptor antagonists (Figure 1B). These experiments show that mPFC layer 1 interneurons receive fast cholinergic

mEPSCs, and they confirm our original conclusion on synaptic release of acetylcholine.

Figure 1: Layer 1 interneurons receive cholinergic miniature Excitatory PostSynaptic Currents (mEPSCs).

A: Example traces of mEPSCs (grey) and average current trace (black). (bath solution contained 1 μ M TTX, 10 μ M DNQX, 25 μ M AP5, 10 μ M gabazine, 4 mM calcium. The intracellular solution contained CsCl).

B: The number of mEPSCs observed in two minute time bins is rapidly reduced to zero when antagonists (MLA and DH β E) of nicotinic Acetylcholine receptors (nAChRs) are applied (n=5 of 5 recorded neurons).

C: left: Amplitude distribution of all recorded mEPSCs (n=5 neurons). Despite the modest number of mEPSCs (n=67 events) recorded in total, the histogram might suggest multiple amplitude peaks (quanta). Right: average mEPSC amplitude.

Since these data do not allow any conclusions on the cellular source of the mEPSCs or co-release of ACh and GABA from ChAT-VIP neurons, the figure does not fit the present manuscript.

2. I continue to think that it is essential to specify the precise recording temperature (mean value, range), rather than specifying the approximate value.

Reply: We agree with the reviewer and have now added the temperature range in the methods section.

Apart from these points, this is a nice paper that definitely should be published in Nature Communications.

Reviewer #3 (Remarks to the Author):

This revised version of the manuscript partly (and sometimes superficially) addressed the points I had raised. The finding that mPFC ChAT-VIP interneurons are involved at different phases of attention performance than BF ChAT neurons is interesting. Their specific connectivity patterns as well. This paper is not attempting to link both. How the local cholinergic connectivity scheme relates to attention is not even discussed. For example, the late involvement of the mPFC ChAT-VIP neurons suggests that their inputs or their synapses may display some form of plasticity during tasks requiring long attention. This would have been nice to study, or at least to discuss.

Reply: We agree with the reviewer and have added a sentence to the discussion to highlight this point.

Points that were only partly addressed:

-axonal morphology of VIP-ChAT cells was only superficially addressed with an example picture of EYFP but no cell axonal morphology reconstruction.

Reply: As requested, we have now added images of biocytin-labeled ChAT-VIP neurons with axons from mouse mPFC to Supplementary Figure 1. From these images it is clearly visible that axons of ChAT-VIP neurons abundantly spread to layer 1 and to deep cortical layers 5 and 6, as we show in Figure 4A and as was beautifully described and extensively quantified by Prönneke et al., Cereb Cortex 2015. In addition, the images exemplify the reported bipolar and multipolar soma-dendritic morphologies of ChAT-VIP neurons as reported extensively in the literature (Eckenstein and Thoenen, 1983, 1984; Von Engelhardt et al., 2007), and as shown in Figure 2A of the present manuscript.

- the criteria to identify a significant response are not explained in detail; the authors mainly chose to quantify average delays.

Reply: We thank the reviewer for pointing this unclarity out. The criteria are specified in the Methods section in the paragraph on Quantification and Statistical Analysis: "If the

amplitude of the response was lower than 4 times the standard deviation of the baseline preceding the stimulation', the event was counted as a failure."

In addition to quantifying average delays, we show in Figure 1E and 1H histograms of individual synaptic events, revealing the full distribution of synaptic delays.

- Amplitude of depolarization evoked onto L6 pyramids: I understand that neurons displaying a high resistance can amplify their membrane depolarization in response to small input currents. I was just asking for some quantification of the recorded EPSP.

Reply: Apologies for the misunderstanding. The quantification of the recorded EPSPs are provided in Figure 4D, left panel.

Additional minor points:

VIP-ChAT: from supplementary Figure 1C it does not seem that all EYFP expressing neurons also expressed VIP. Therefore, I feel that the term VIP-ChAT is somehow misleading; this simplified terminology should be better stated and acknowledged.

Reply: ChAT-VIP cells can express VIP and ChAT at different levels. Antibodies can fail to detect low abundant protein, but absence of antibody staining does not necessarily mean complete absence of VIP protein. To illustrate this point, Prönneke et al., 2015 observed in Vip-ires-cre mice crossed with homozygous Ai9 mice introducing a cre-dependent TdTomato reporter, that the VIP antibody staining did not 100% overlap with TdTomato, so some VIP cells (called by the TdTomato expression) were not VIP positive when judged from the antibody staining (see Figure 2A and 2A' in Prönneke et al., 2015). We have added this information to the legend of supplementary Figure 1C.

Lines 129-132 : this sentence reads quite awkwardly, I think it should be placed elsewhere, maybe in the introduction or at the beginning of the results.

Reply: The sentence has now been altered.

Line 152: "than" instead of "then"

Reply: Corrected

Reviewers' Comments:

Reviewer #2:

Remarks to the Author:

This is a nice paper that definitely should be published in Nature Communications.

The miniature EPSCs are nice and important data. They could be easily included (and from my point of view must be included) in the Supplementary Material section.

Dear Editor,

We have addressed the remaining point raised by Reviewer 2.

Reviewer #2 (Remarks to the Author):

This is a nice paper that definitely should be published in Nature Communications.

The miniature EPSCs are nice and important data. They could be easily included (and from my point of view must be included) in the Supplementary Material section.

Author reply: As requested, we have added the figure to the Supplementary document and it now features as Supplementary Figure 1. We also refer to this figure in the text of the results section on page 5.

Best regards, on behalf of the authors,
Huib Mansvelder